EMBO
Molecular Medicine

# Clusterin knockdown sensitizes prostate cancer cells to taxane by modulating mitosis

Nader Al Nakouzi[1], Chris Kedong Wang[1], Eliana Beraldi[1], Wolfgang Jager[1], Susan Ettinger[1], Ladan Fazli[1], Lucia Nappi[1], Jennifer Bishop[1], Fan Zhang[1], Anne Chauchereau[2,3], Yohann Loriot[2,3] & Martin Gleave[1,*]

## Abstract

Clusterin (CLU) is a stress-activated molecular chaperone that confers treatment resistance to taxanes when highly expressed. While CLU inhibition potentiates activity of taxanes and other anti-cancer therapies in preclinical models, progression to treatment-resistant disease still occurs implicating additional compensatory survival mechanisms. Taxanes are believed to selectively target cells in mitosis, a complex mechanism controlled in part by balancing antagonistic roles of Cdc25C and Wee1 in mitosis progression. Our data indicate that CLU silencing induces a constitutive activation of Cdc25C, which delays mitotic exit and hence sensitizes cancer cells to mitotic-targeting agents such as taxanes. Unchecked Cdc25C activation leads to mitotic catastrophe and cell death unless cells up-regulate protective mechanisms mediated through the cell cycle regulators Wee1 and Cdk1. In this study, we show that CLU silencing induces a constitutive activation of Cdc25C via the phosphatase PP2A leading to relief of negative feedback inhibition and activation of Wee1-Cdk1 to promote survival and limit therapeutic efficacy. Simultaneous inhibition of CLU-regulated cell cycle effector Wee1 may improve synergistic responses of biologically rational combinatorial regimens using taxanes and CLU inhibitors.

**Keywords** cabazitaxel; Cdc25C; clusterin; mitotic exit; Wee1
**Subject Categories** Cancer; Pharmacology & Drug Discovery; Urogenital System

## Introduction

Prostate cancer (PCa) is the most commonly diagnosed cancer in North American men and the second leading cause of cancer-related death (Cancer Facts & Figures 2015. American Cancer Society). PCa is initially responsive to hormone therapy, but inevitably progresses to lethal, castrate-resistant PCa (CRPC) (Gleave *et al*, 1998, 2001). The potent androgen receptor (AR) pathway inhibitors, abiraterone and enzalutamide, as well as the taxanes, docetaxel and cabazitaxel, all prolong survival in CRPC. Docetaxel is the first approved agent outside of hormonal therapy with a demonstrated survival benefit in CRPC (Petrylak *et al*, 2004; Tannock *et al*, 2004), while cabazitaxel is approved for docetaxel-recurrent metastatic CRPC (Yap *et al*, 2012). Taxanes function by stabilizing the dynamic polymerization of microtubules resulting in inhibition of mitosis and cell death. The ability of microtubules to assemble and disassemble is critical for mitosis and thus targeting microtubules preferentially targets rapidly dividing cancer cells. Taxanes can also affect AR signaling through alteration of microtubule-associated AR cellular transport and nuclear translocation (Zhu *et al*, 2010). Moreover, cabazitaxel remains active after abiraterone or enzalutamide treatment (Al Nakouzi *et al*, 2014).

Considerable efforts have focused on defining the molecular mechanisms that lead to taxane resistance, but few have been successful in the clinical setting. One example involves the molecular chaperone, clusterin (CLU), which is expressed in many human cancers, including PCa, where it increases following castration to become highly expressed in CRPC (Miyake *et al*, 2000; July *et al*, 2002). While CLU inhibition potentiates the activity of many anti-cancer therapies in preclinical models (Miyake *et al*, 2005a,b; Zoubeidi *et al*, 2010; Chun, 2014), the CLU inhibitor custirsen (OGX-011) failed to prolong survival when combined with docetaxel in CRPC (Chi *et al*, 2015). These results suggest that compensatory survival mechanisms may occur when CLU is inhibited in PCa cells, especially when used in combination with taxanes.

Dysregulated mitosis may represent one mechanism enabling treatment resistance to taxanes and other anti-cancer drugs. While CLU has been linked to control of cell cycle by interacting directly with the maturation-promoting factor [MPF; Cdk1/cyclin B1] and cyclin B1 proteolysis (Scaltriti *et al*, 2004), its role in mitotic regulation is poorly defined. In eukaryotic cells, an active MPF promotes entry into mitosis. Prior to mitosis, MPF kinase activity is inhibited by Wee1 kinase-mediated phosphorylation of the Cdk1 catalytic subunit on two residues, T14 and Y15 (Nurse, 1994).

1   The Vancouver Prostate Centre and Department of Urologic Sciences, University of British Columbia, Vancouver, BC, Canada
2   Department of Cancer Medicine, Gustave Roussy, Cancer Campus, Grand Paris, University of Paris-Sud, Villejuif, France
3   INSERM U981, Villejuif, France
    *Corresponding author. Tel: +1 604 875 4818; Fax: +1 604 875 5654; E-mail: m.gleave@ubc.ca

Dephosphorylation of Cdk1 and subsequent entry into mitosis are catalyzed by Cdc25C, which in turn is regulated by cell cycle-dependent phosphorylation events (Gautier *et al*, 1991; Kumagai & Dunphy, 1991). Hyperphosphorylation of Cdc25C during mitosis is thought to stimulate its phosphatase activity (Hoffmann *et al*, 1993). Premature activation of Cdc25C is prevented by phosphorylation of specific residues in Cdc25C during interphase that are distinct from the sites phosphorylated in M phase (Ogg *et al*, 1994; Peng *et al*, 1997; Dalal *et al*, 1999). At G2/M, Cdc25C localizes in the nucleus in coordination with cyclin B1 (Toyoshima-Morimoto *et al*, 2002) and positive regulatory sites including threonine 48 (T48) are phosphorylated. Mutation of T48 to alanine prevents Xenopus Cdc25C activation *in vitro* (Pines & Hunter, 1992). In addition, T48 can be phosphorylated by MAP kinase ERK2 (Wang *et al*, 2007), although factors regulating its dephosphorylation are not fully understood. PP2A is a phosphatase and a negative regulator of Cdc25C at mitosis. Failure of PP2A to dephosphorylate Cdc25C at mitosis results in prolonged hyperphosphorylation and activation of Cdc25C, causing persistent dephosphorylation and, hence, activation of Cdk1. This constitutive activation of Cdc25C and Cdk1 leads to a delayed exit from mitosis (Forester *et al*, 2007).

Cytotoxic drugs like taxanes are believed to selectively target cells in mitosis; quiescent cells or cycling cells that do not reach mitosis during drug exposure are spared (Baguley *et al*, 1995; Orth *et al*, 2011). Therefore, sensitizing cells to taxane, in particular after CLU inhibition, might occur through mitosis regulation. Little is known about how CLU affects these complex regulators of mitosis and hence taxane resistance. In this paper, we explored the biological effects of CLU on mitosis and define a mechanism whereby CLU silencing induces constitutive activation of Cdc25C and mitotic exit delay that enhances taxane sensitivity in prostate cancer cells.

## Results

### CLU knockdown modulates expression of mitosis regulators

As previously reported by our group and others, down-regulation of CLU reduces cell proliferation (Trougakos *et al*, 2004; Niu *et al*, 2012; Wang *et al*, 2014). In PC3 cells, down-regulation of CLU results in decreased cell proliferation (Fig 1A) and a 40% mean increase in population of G2/M cells (Fig 1B). To further investigate the role of CLU in cell cycle regulation, microarray transcriptome analysis was performed on PC3 cells transiently transfected with siRNA targeting clusterin mRNA (siCLU) or scramble RNA (siSCR). As shown in Fig EV1A, differential expression profiling identified many biologically related gene clusters involved in the regulation of apoptosis, cell cycle progression and cell growth/proliferation (GEO number: GSE67256). Among the cell cycle regulators, CLU knockdown resulted in increased mRNA expression of Cdc25C phosphatase, Cdk1 kinase (Cdk1) and Wee1 kinase (Fig 1C left panel). As expected, proteins levels of Cdc25C, Cdk1 and Wee1 were also up-regulated when CLU was silenced (Fig 1C right panel). These proteins regulate cell cycle progression both in to, and exit from, mitosis. Since regulation of the activity of these proteins during cell cycle is precisely controlled by phosphorylation and dephosphorylation events, we conducted a phosphokinome analysis in LNCaP cells (Fig 1C left panel) and found that the phosphorylation patterns of

these proteins, as well as several others, are indeed altered when CLU is knocked down. In total, 4 genes were actually up-regulated at both mRNA and protein levels after CLU silencing as shown by the diagram in Fig 1C (central panel). Collectively, these data suggest that CLU affects cell cycle progression by altering the expression levels of key modulators of mitosis duration.

### Clusterin silencing leads to increased levels and activation of Cdc25C in human cancer cells

To investigate functional links between CLU and Cdc25C, we first evaluated the expression level of Cdc25C in different PCa cell lines (LNCaP, VCaP, and PC3) after CLU silencing. As shown in Fig 2A and B, basal levels of Cdc25C mRNA and protein expression are up-regulated after CLU silencing. Re-expression of CLU in PC3 cells rescued the basal level of Cdc25C, confirming an inverse correlation between CLU and Cdc25C (Fig 2C). Next, changes in Cdc25C protein levels were also evaluated in LNCaP xenografts from mice treated with OGX-011, a CLU antisense drug. Consistent with *in vitro* results, Cdc25C protein levels were increased in LNCaP tumors treated with OGX-011 (Fig 3A). Moreover, when LNCaP xenografts were categorized into high or low CLU expressing tumors, Cdc25C negatively correlated with CLU at both protein and mRNA levels (Fig 3B). These *in vitro* and *in vivo* findings were further corroborated by *in silico* analysis using GeneSapiens (Kilpinen *et al*, 2008), a collection of re-annotated and renormalized Affymetrix microarray experiments collected from various publicly available sources (Fig EV1B). In particular, transcript levels of Cdc25C negatively correlated with CLU expression across 460 prostate adenocarcinoma human tissue samples ($P < 0.001$; $r = -0.23$). Finally, this inverse correlation between Cdc25C and CLU levels was further investigated by immunohistochemistry in a cohort of radical prostatectomy specimens from patients who were either untreated, treated with 3 months of neoadjuvant hormone therapy, or treated with 3 months neoadjuvant hormone therapy with the addition of OGX-011 (Chi *et al*, 2005) (Fig 3C). As previously demonstrated, CLU increases after NHT and is decreased after OGX-011, significantly. In addition, there was a trend in Cdc25C increase after OGX-011, although statistical significance was not reached probably due to the low sample size of the study. Overall, these data indicate a strong negative correlation between levels of CLU and the key regulator of mitosis, Cdc25C.

### Clusterin binds to Cdc25C

As a secreted chaperone, CLU interacts with proteins to stabilize and facilitate their interactions with others (Poon *et al*, 2002). To investigate whether CLU binds Cdc25C, PC3 cells were transfected with plasmids encoding Cdc25C and CLU. Co-immunoprecipitation using Cdc25C antibody reveals interaction with CLU, and this was confirmed with the reverse IP using CLU (Fig 4A). A similar interaction was detected by co-immunoprecipitating the endogenous proteins (Fig EV1C). Furthermore, immunofluorescence staining for CLU and Cdc25C was performed in PC3 cells, and Fig 4B clearly shows co-localization of these two proteins. Moreover, this interaction was further confirmed by proximity ligation assay. Red dots in Fig 4C indicate interaction between CLU and Cdc25C in siSCR-treated cells, and this signal was lost when CLU was silenced. CLU and Cdc25C

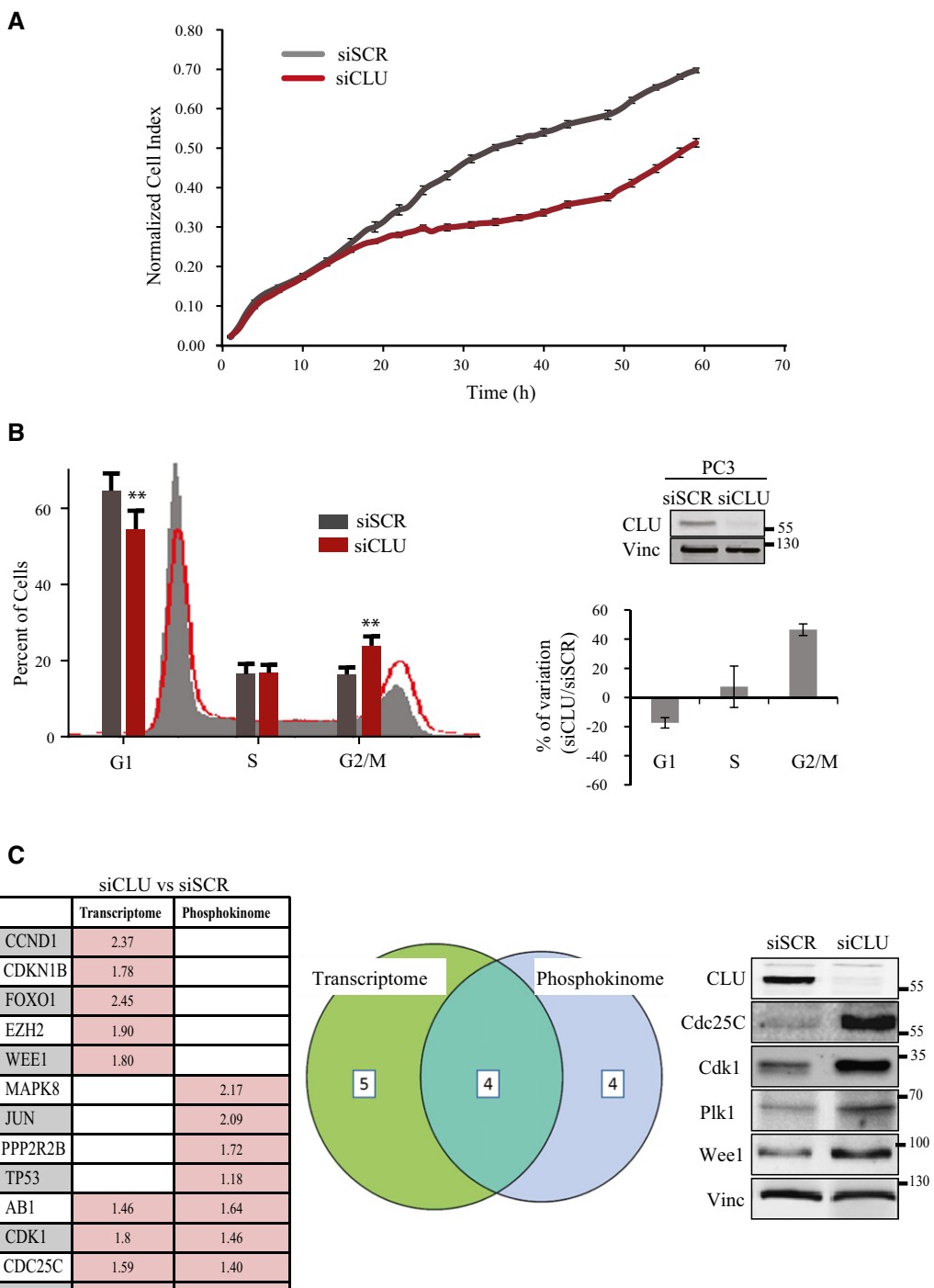

**Figure 1.  CLU is required for proliferation and cycling of PC3 cells.**

A   Cell proliferation for PC3 cells transfected with siSCR or siCLU monitored in real time for 60 h with SP Dynamic xCELLigence system (Voltage 2.5 K). Error bars represent mean ± SEM, *n* = 4. Slopes compared by ANCOVA test, *P* < 0.0001.

B   Left panel: Cell cycle profile of PC3 cells transfected with siSCR or siCLU. Right panel: Percentage quantification of variation in G1, S, and G2/M phases. Error bars represent mean ± SEM, *n* = 3, **P* < 0.01 by paired Student's *t*-test (G2/M population *P* = 0.0058; G1 *P* = 0.008). Inset: Western blot with CLU antibody in PC3 cells after CLU silencing. Vinculin was used as loading control.

C   Left panel: RNA microarray analysis comparing PC3 cells transfected with siCLU versus siSCR, and Kinexus phosphokinome microarray analysis comparing LNCaP cells transfected with siCLU versus siSCR. Target genes expression is shown as fold change relative to siSCR, *P* ≤ 0.05 by an unpaired *t*-test for all the genes listed. Central panel: Venn diagram to graphically illustrate the overlap of mRNA and proteins involved in the regulation of mitosis in both transcriptome and phosphokinome analysis. The comparison of these two experiments shows modification of 4 mitosis regulators. Right panel: Western blot validating the expression levels of target proteins.

## A

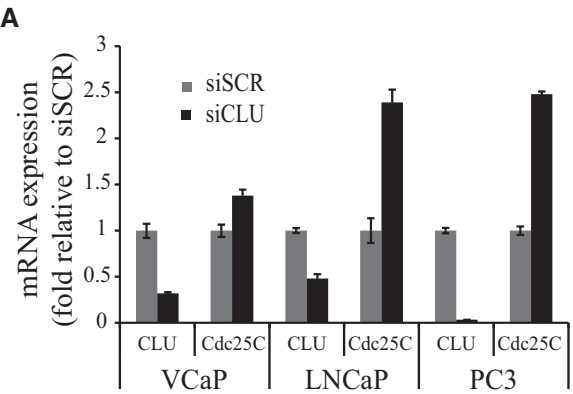

## B

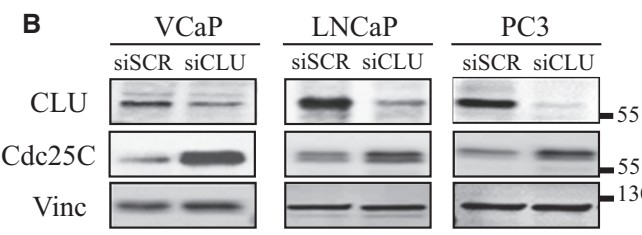

## C

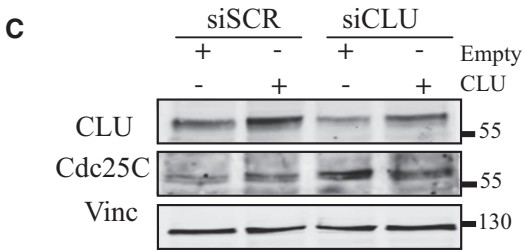

**Figure 2.  CLU modulates Cdc25C expression *in vitro*.**

A   Real-time PCR for CLU and Cdc25C in three different prostate cancer cell lines. Target genes expression was calculated relative to GAPDH and normalized to siSCR. Error bars represent mean ± SD, $n = 3$.

B   Western blot for CLU and Cdc25C after siSCR or siCLU transfection in the same prostate cancer cell lines as in A. Vinculin was used as loading control.

C   Western blot for CLU and Cdc25C in PC3 cells after transfection with siSCR or siCLU followed by transfection with CLU or empty expression plasmid for an additional 48 h. Vinculin was used as loading control.

PLA specificity was also assessed by incubation with antibodies of non-related proteins (Fig EV1D). Overall, these data represent a strong evidence of the interaction between CLU and Cdc25C.

## Clusterin silencing leads to a delay in exit from mitosis

Based on the above observations, we postulated that CLU could be involved in regulation of cell exit from mitosis. To address this possibility, we performed cell cycle analysis in PC3 cells after CLU knockdown and synchronization to M phase using "thymidine/ nocodazole block". Cell cycle analysis shows that after nocodazole release, control cells were able to switch from M phase to G1 phase after 2 h, while CLU-silenced cells were trapped in mitosis (Fig 5A). As shown by immunofluorescence after nocodazole treatment, CLU-silenced PC3 cells remain in mitosis for more than 5 h, compared to control cells that divided after 3 h of release (Fig 5B). Moreover, mitotic and segregation abnormalities (e.g., frequent multipolar spindles, lagging chromosomes) were observed (Fig 5B lower panel, 120, 180, and 300 min), abnormalities known to induce the mitotic checkpoint and lead to mitotic catastrophe. To confirm that this phenomenon was indeed a delay in mitosis, cells were stained for phosphohistone H3, which marks activation of Cdk1 during mitosis, and examined at various time points after release using FACS analysis. Figure 5C illustrates that the normal post-mitotic decrease in phospho-Ser-10 histone H3, observed in control cells, was significantly delayed in CLU-silenced cells, indicating prolonged retention of cells in mitosis. The same trend was confirmed in asynchronized cells, where the percentage of the cells positive for phosphohistone H3 shifted from 0.8% to 10% after treatment with siCLU compared to siSCR (Fig EV1E).

Having found an inverse correlation between CLU and Cdc25C, we next tested the hypothesis that phosphorylation events regulating Cdc25C and Cdk1 activity were altered by CLU knockdown. For cells to enter mitosis, Cdc25C must be activated by hyperphosphorylation at the amino terminus (including, but not limited to, phosphorylation of T48), which causes a marked electrophoretic mobility shift and is a well-established marker of Cdc25C activation (Strausfeld *et al*, 1994; Forester *et al*, 2007). Subsequently, Cdk1 is phosphorylated at an inhibitory site, leading to degradation of cyclin B1 and exit from mitosis (Clute & Pines, 1999; Wolf *et al*, 2006; Sullivan *et al*, 2008). PC3 cells treated with siSCR or siCLU were synchronized with nocodazole, and protein levels of three main regulators of mitosis exit, Cdc25C, Cdk1, and cyclin B1, were evaluated at different time points after release. Interestingly, we found that control cells exit mitosis within 2–3 h, marked by the loss of the active hyperphosphorylated form of Cdc25C and the return to its 60-kDa interphase, inactive form. In addition, Cdk1 inhibitory phosphorylation appears at 3 and 5 h after release, followed by cyclin B1 degradation (Fig 5D). Conversely, CLU-knockdown (KD) cells, which exit mitosis later (Fig 5A), showed a slower

**Figure 3.  Clusterin and Cdc25C are inversely correlated *in vivo* and in human patients.**

A   Upper panel: Western blot for CLU and Cdc25C from LNCaP xenografts harvested from mice treated with OGX-011 ($n = 8$) or SCR ASO control ($n = 6$). Vinculin was used as loading control. Lower panel: Quantification of the protein levels, relative to the loading control, reported as dot plots. *$P < 0.05$ by $t$-test followed by Welch's correction (CLU $P = 0.049$, Cdc25C $P = 0.024$).

B   Western blot with the correspondent quantification of protein levels (left) and real-time PCR (right) from untreated LNCaP xenografts harvested at endpoint. The negative correlation between CLU-RNA and Cdc25C-RNA $\Delta\Delta C_T$ values was calculated using a spearman test ($\rho = -0.86$; $P = 0.0107$).

C   Immunohistochemical staining for CLU and Cdc25C of radical prostatectomy samples from historical control specimens with no neoadjuvant therapy ($n = 5$) or neoadjuvant hormone therapy (NHT) ($n = 5$) or the combination of NHT and OGX-011 ($n = 9$) for 3 months prior to surgery. Average score staining from the three groups (left) and representative sections for CLU and Cdc25C (right). Error bars represent mean ± SEM. CLU: *$P < 0.05$; untreated versus 3mNHT ($P = 0.0213$); 3mNHT versus 3mNHT + OGX-011 ($P = 0.012$) by ANOVA followed by Bonferroni's *post hoc* analysis. All cases were normalized by clinical stage, Gleason score, and serum PSA. For each patient, the pathologist selected the area with the highest Gleason score. Scale bar represents 100 μm.

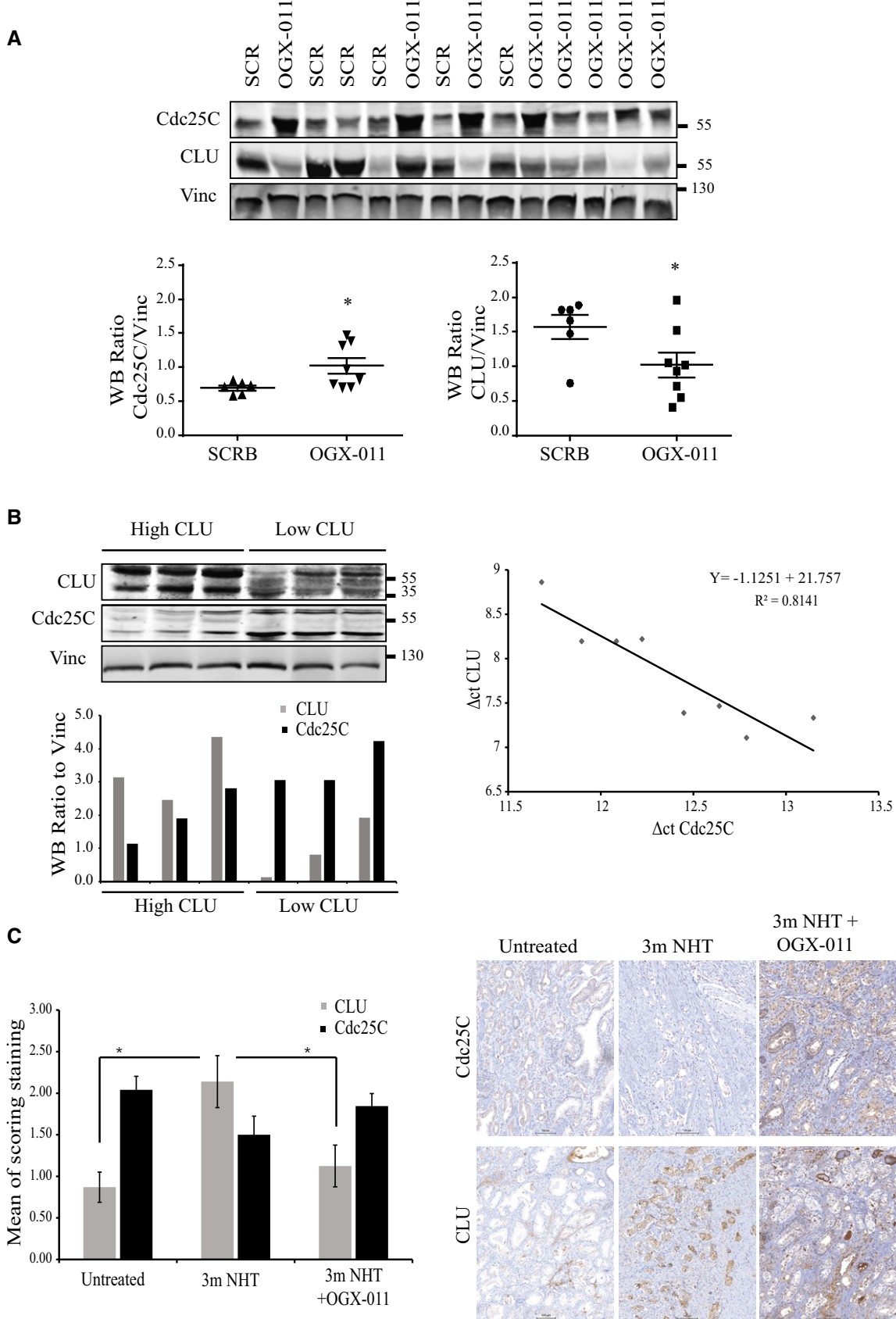

**Figure 3.**

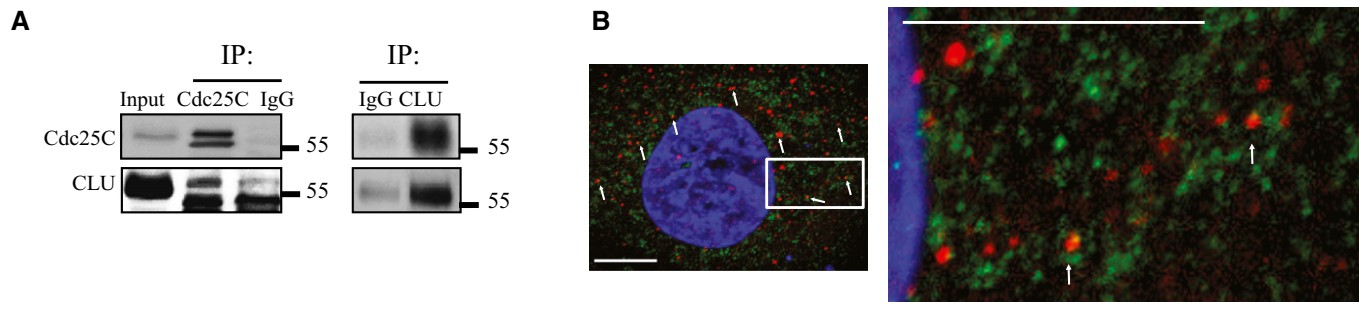

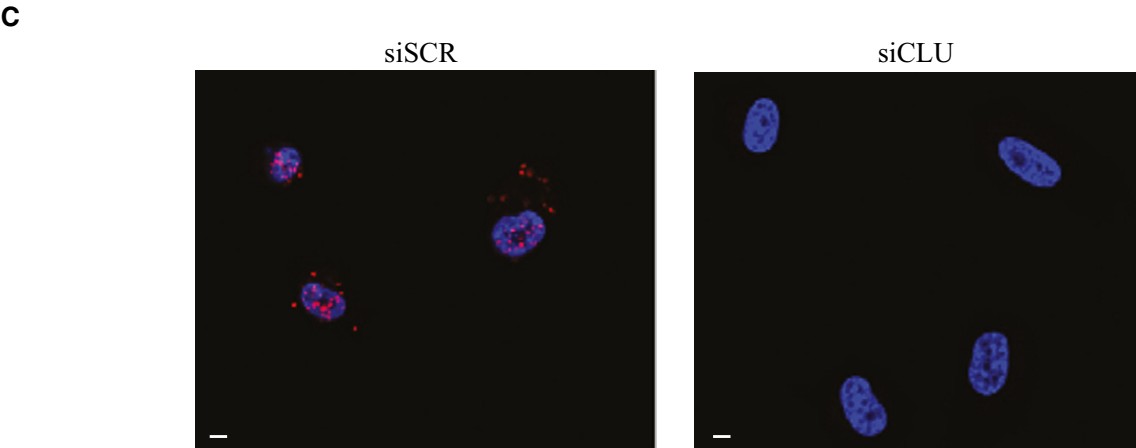

**Figure 4. Analysis of Cdc25C/CLU complex in PC3 cells by co-immunoprecipitation, immunofluorescence microscopy, and *in situ* proximity ligation assay (PLA).**

A   Co-immunoprecipitation from parental PC3 cells transfected with CLU and Cdc25C plasmid expression vectors. Proteins were immunoprecipitated with specific anti-Cdc25C (left) and CLU (right) antibodies and analyzed by immunoblotting.

B   Dual immunofluorescence staining with Cdc25C (green) and CLU (red) antibodies in PC3 cells. Confocal microscopy was used to identify the interaction (white arrows). DNA was counterstained with (DAPI) (blue). Scale bar represents 10 μm.

C   Duolink proximity ligation assay between Cdc25C and CLU in PC3 cells. Confocal microscopy was used to detect the interaction (red dots). DNA was counterstained with DAPI (blue). PC3 cells transfected with siCLU were used as a negative control. Scale bar represents 10 μm.

dephosphorylation of Cdc25C, with persistent Cdc25C activation at 3 and 5 h after nocodazole release and an absence of Cdk1 phosphorylation at 3–5 h (Fig 5D). Moreover, cyclin B1 expression levels are still present, indicating persistent activity of MPF. This unchanged Cdk1 activity is consistent with the presence of histone H3 phosphorylation at similar time points (Fig 5C). To determine which phospho-site of Cdc25C is affected by CLU-KD, a panel of 3 phospho-Cdc25C antibodies was analyzed under the same conditions. As indicated in Fig 6A, the levels of Cdc25C-T48, detected by Western blotting and immunofluorescence, were predominantly up-regulated

after CLU-KD. Thus, these data show that loss of CLU induces accumulation of the phosphorylated active form of Cdc25C, leading to constitutive activation of Cdk1 and mitotic exit delay.

**CLU-KD induces Cdc25C phospho-T48 accumulation reducing PP2A phosphatase activity**

We previously demonstrated that delay in mitosis exit in PC3 cells after CLU-KD correlated with an accumulation of Cdc25C-T48. To define cellular pathways involved in this process, we investigated

**Figure 5. Knockdown of CLU leads to delay in the exit from mitosis.**

A   Flow cytometry for PC3 cells after transfection with siSCR or siCLU and synchronization to the M phase with thymidine/nocodazole block. DNA content was analyzed at the indicated time point after nocodazole release. Histograms show % of cells in G2/M, over 5 h period. Error bars represent mean ± SEM, *n* = 3, *P < 0.05, ****P < 0.0001 by ANOVA followed by Bonferroni's *post hoc* correction. (2 h P = 0.0142).

B   Immunofluorescence in PC3 cells after siSCR or siCLU transfection, synchronization and nocodazole release at indicated time points. Mitotic spindle formation was assessed by confocal microscopy, using pericentrin (centrioles; red) and β-tubulin (microtubules; green) antibodies. DNA was counterstained with (DAPI) (blue). White arrows represent segregation abnormalities. Scale bar represents 10 μm.

C   Flow cytometry to assess phosphohistone H3 levels in the same setting as in (A). Error bar represents mean ± SEM, *n* = 3, ****P < 0.0001 by ANOVA followed by Bonferroni's *post hoc* correction.

D   Western blot for CLU, Cdc25C, Cdk1, and cyclin B1 antibodies in PC3 cells after transfection with siSCR or siCLU, synchronization and nocodazole release at the indicated time. Vinculin was used as loading control.

**A**

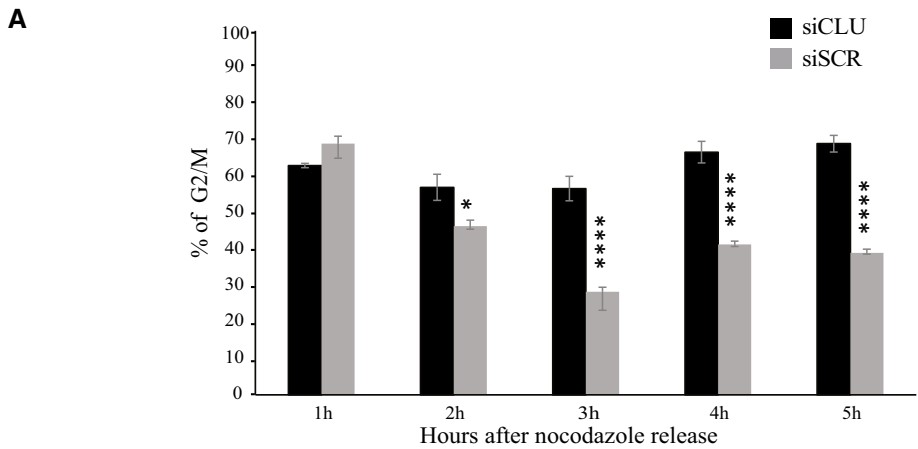

**B**

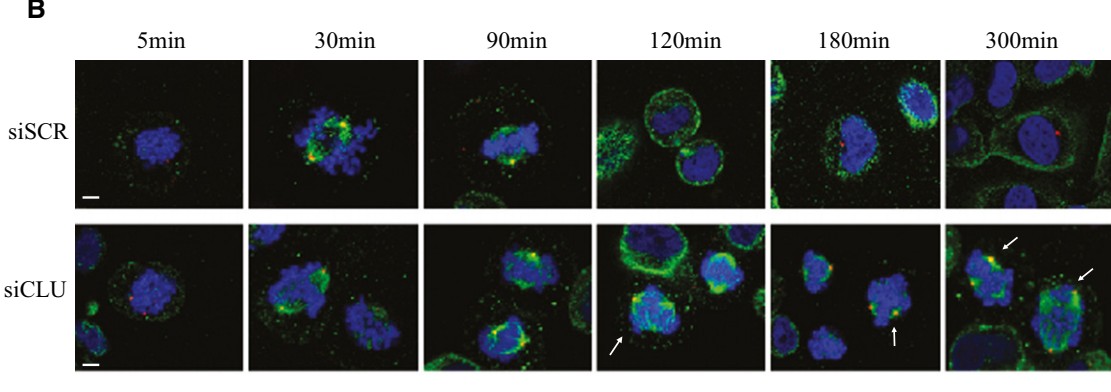

**C**

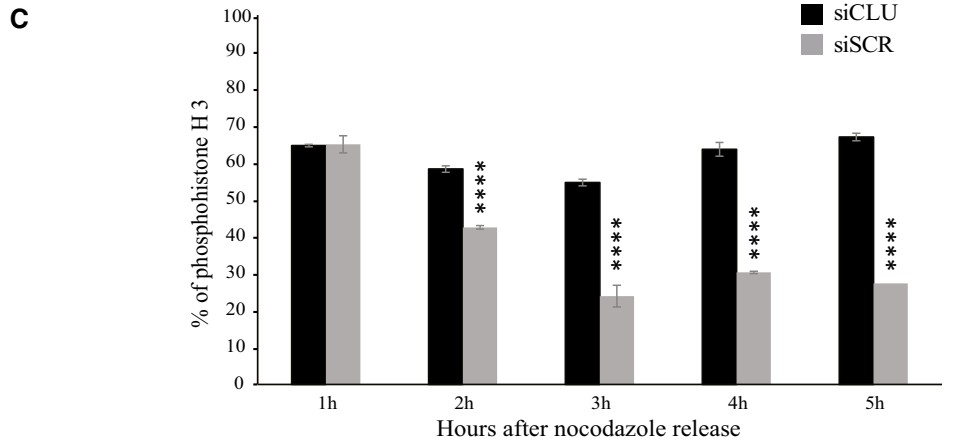

**D**

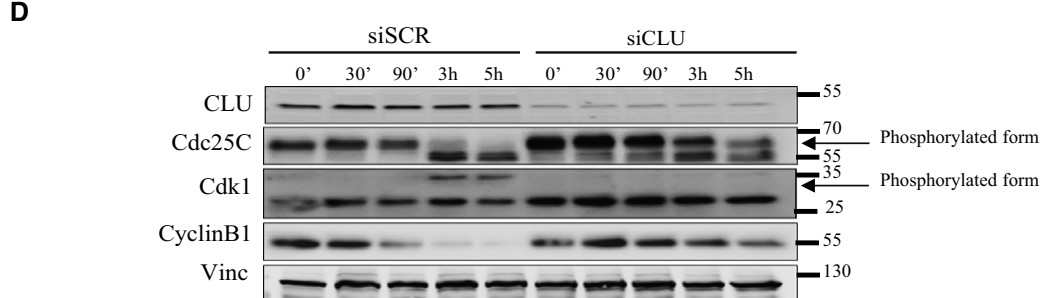

**Figure 5.**

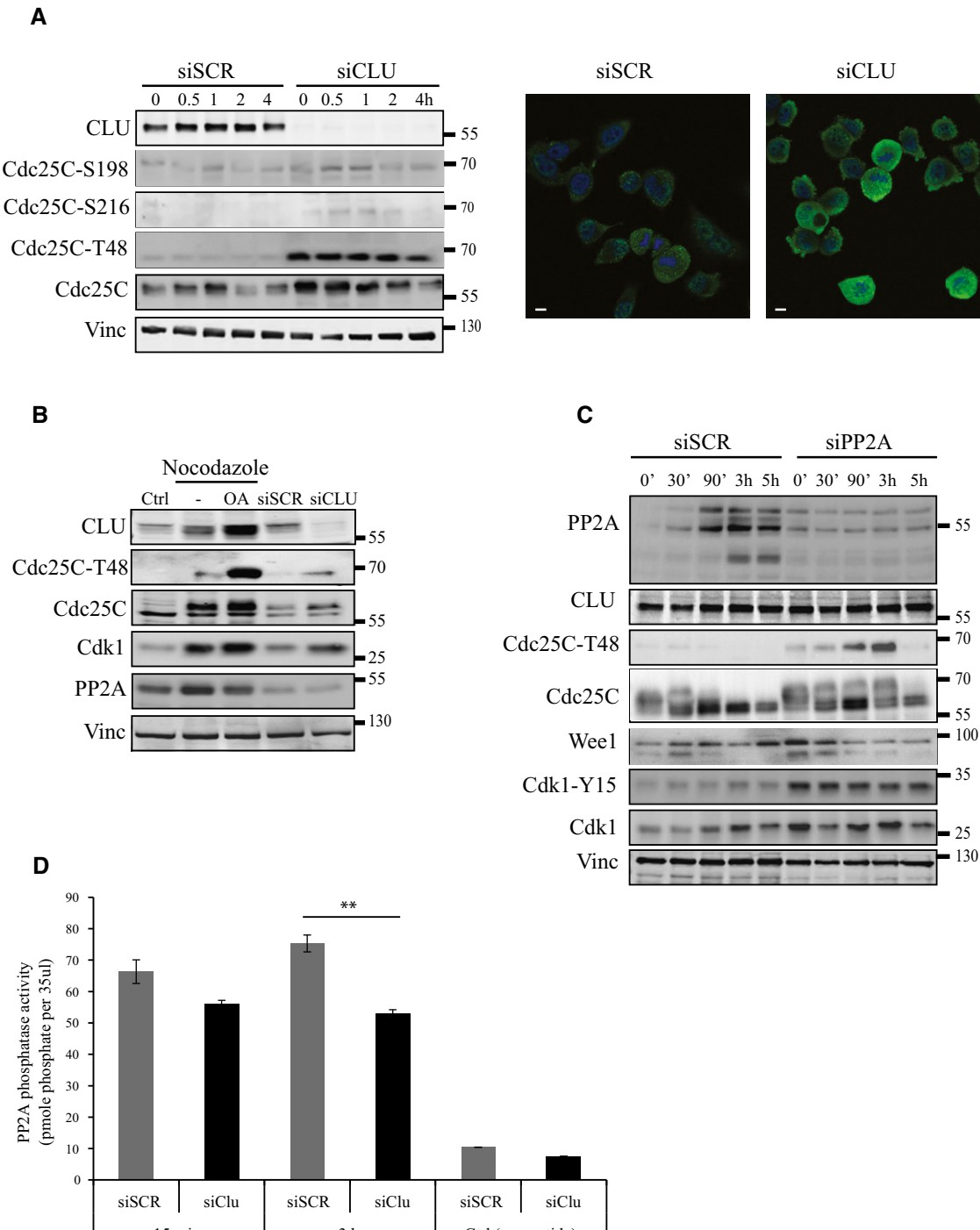

**Figure 6.  Decreased PP2A activity causes Cdc25C-T48 accumulation after CLU silencing.**

A   Left panel: Western blot for CLU, Cdc25C, Cdc25C-T48, Cdc25C S216, and Cdc25C S198 in PC3 cells after siSCR or siCLU transfection, synchronization and nocodazole release at indicated time points. Vinculin was used as loading control. Right panel: Immunofluorescence for Cdc25C-T48 in PC3 cells after siSCR or siCLU transfection. DNA was counterstained with (DAPI) (blue). Scale bar represents 10 μm.

B   Western blot for CLU, Cdc25C-T48, Cdk1, and PP2A in PC3 cells after siSCR or siCLU transfection and after synchronization followed by treatment with or without okadaic acid (OA) (100 nM) for 12 h. Proteins were extracted 15 min after nocodazole release. Vinculin was used as loading control.

C   Western blot for PP2A, CLU, Cdc25C-T48, Cdc25C, Wee1, and Cdk1 in PC3 cells transfected with siSCR or siPP2A synchronized and nocodazole released at indicated time points. Vinculin was used as loading control.

D   PP2A phosphatase activity measurement in PC3 cells after siSCR or siCLU transfection, synchronization and nocodazole release at indicated time points. Error bar represents mean ± SEM, n = 3, **P < 0.01 (P = 0.0025) by ANOVA followed by Bonferroni's *post hoc* comparisons. Cells treated with no peptide were used as a negative control.

whether this results from increased T48 accumulation, via ERK/MAP kinase pathway, or decreased T48 dephosphorylation due to decreased PP2A phosphatase activity. PC3 cells were treated with siCLU or siSCR, synchronized to M phase using "thymidine/nocodazole block" and analyzed after release for assessment of ERK/MAPK activity. While ERK phosphorylation appears to be reduced after CLU-KD, Cdc25C-T48 is highly accumulated (Fig EV1F, left panel). Moreover, when MAPK activity is inhibited chemically using UO126, CLU-KD-associated T48 is still accumulating compared to siSCR (Fig EV1F, right panel), suggesting that increased T48 did not result from ERK/MAP kinase activation. Next, to explore whether phosphatase PP2A affected T48 phosphorylation levels, PC3 cells were synchronized in M phase in the presence or absence of the specific PP2A inhibitor, okadaic acid, and collected for Western blot after 30 min of release. Figure 6B shows a higher accumulation of Cdc25C-T48, Cdc25C, and Cdk1 in the presence of okadaic acid with unchanged PP2A phosphatase levels, similar to siCLU condition. Same results were observed when PP2A was silenced using specific siPP2A (Fig 6C). In addition, Cdk1-Y15 and Wee1 were accumulated after PP2A silencing while CLU levels remain unchanged. Furthermore, PP2A activity was significantly reduced after CLU silencing specifically 3 hours after nocodazole release (Fig 6D). Collectively, these results suggest that T48 accumulation after CLU-KD is not due to an increase of MAPK pathway but rather to lower PP2A phosphatase activity.

## Activation of Cdc25C by CLU-KD is compensated by Wee1 up-regulation

Based on the above observations, CLU-KD results in activated Cdc25C that should cause constitutive activation of Cdk1 and ultimately provoke mitotic catastrophe and apoptosis (Castedo *et al*, 2002; Niida *et al*, 2005; Forester *et al*, 2007). Interestingly, CLU-KD cells continue to grow and divide (Fig 1A). To better understand the regulation of Cdk1 in this context, we next examined the phosphorylation status of Cdk1 at different times in PC3 cells treated with siSCR or siCLU and released after nocodazole treatment. Unexpectedly CLU-KD shows an increase of Cdk1 phosphorylated on Y15 associated with a parallel increase of Wee1 kinase protein levels (Fig 7A). A similar increase of Wee1 was confirmed at mRNA level by quantitative PCR analysis (Fig 7B). Moreover, inhibition of Wee1, using a specific inhibitor (MK-1775), further increased phosphohistone H3 cell population in the absence of CLU (Fig 7C) and abrogated the CLU-KD associated accumulation of Cdk1 and Cdk1-Y15 protein levels (Fig 7D). Overall, these results indicate that CLU silencing increases Cdk1 activity through Cdc25C, but leads to relief of feedback inhibition via Wee1 up-regulation, a compensatory survival response adopted by PC3 cells under conditions of CLU-KD.

## Clusterin silencing sensitizes PC3 cells to cabazitaxel

Since cabazitaxel is approved for the treatment of docetaxel-pretreated CRPC patients, we next set out to determine whether CLU-KD could enhance the activity of this drug. We first examined the activity of cabazitaxel in the presence and absence of CLU by calculating IC50 and showed that either siCLU or OGX-011 reduced cabazitaxel IC50 by 50% in PC3 cells (Fig 8A), as previously shown

for docetaxel (So *et al*, 2005; Sowery *et al*, 2008). It is well known that cabazitaxel induces mitotic arrest followed by apoptotic cell death (Vrignaud *et al*, 2014). To confirm that the enhanced anti-cancer activity of cabazitaxel, combined with CLU silencing, resulted from altered mitotic activity, we quantitated the number of cells displaying mitotic nuclei or multiple nuclei at various doses of cabazitaxel +/− CLU-KD. Under conditions of CLU-KD, 1.5 nM cabazitaxel resulted in a higher number of mitotic cells, and this effect was dose-dependent up to 12.5 nM, where nearly 82% of cells had abnormal or mitotic nuclei. Thus, combining CLU-KD and cabazitaxel resulted in greater anti-mitotic and anti-proliferative activity compared with cabazitaxel alone (Fig 8B). These results show a tight correlation between the anti-mitotic and the cytotoxic effects of the combination. In addition, since taxane effect on survival can occur at different time points, a clonogenic assay was performed and showed lower number of clones after CLU-KD compared to SCR after 15 days (Fig 8C).

## CLU-Cdc25C-Wee1 pathway is a resistance mechanism to cabazitaxel

It is well established that taxanes induce mitotic arrest followed by apoptotic cell death (Hernandez-Vargas *et al*, 2007; Gascoigne & Taylor, 2009). To facilitate the study of CLU's role in mitosis regulation as mechanism of taxane resistance, we used cabazitaxel-resistant PC3 (PC3-R-caba) and IGR-CaP1 (IGR-CaP1-R-caba) cells as previously described (Nakouzi *et al*, 2014). Cell cycle analysis confirmed that the resistant cells, showing a cell cycle distribution similar to the parental cells, were not arrested in M phase after cabazitaxel treatment compared to the parental cells (Fig EV2A). Furthermore, to evaluate rates of mitosis progression, parental and resistant cell lines were synchronized into mitosis using thymidine/nocodazole blockage and released at different time. As shown in Fig 9A, after 1 h of release, 24% of resistant cells exit mitosis compared to 11% in parental cells. The majority of resistant cells reached G1 at 2 h while in parental cells the majority of cells reach G1 after 3 h, indicating that cabazitaxel-resistant cells exit mitosis faster than parental cells. As expected, cabazitaxel-resistant cells displayed increased CLU compared to WT that was associated with lower levels of the mitotic regulators, Cdc25C and Cdk1, suggesting faster progression through mitosis (Fig 9B). In addition, CLU-KD in cabazitaxel-resistant cell lines leads to increased levels of Cdc25C, Cdk1, and Cdc25C-T48, thereby re-establishing a sensitive molecular phenotype (Fig 9B). To further confirm the ability of CLU-KD to resensitize resistant cells to cabazitaxel, PC3-R-caba cells were maintained for three passages without cabazitaxel before treatment with cabazitaxel or cabazitaxel + CLU-KD combination. While 10 nM cabazitaxel has a slight effect on PC3-R-caba, CLU-KD reduces survival by 50%, recapitulating what we have observed in parental PC3 cell lines (Fig 9C). Since we previously hypothesized that Wee1 up-regulation after CLU-KD could represent a possible survival compensatory mechanism in PC3 cells, we next investigated whether this was also seen in PC3-R-caba cells after CLU inhibition. As shown in Fig 9B (right panel), Wee 1 protein levels increase after CLU-KD. Furthermore, when PC3-R-caba cells were treated with MK-1775 in combination with CLU inhibition, a significant decrease of cell survival (Fig 9D), associated with higher level of cleaved PARP (Fig 9E), was observed,

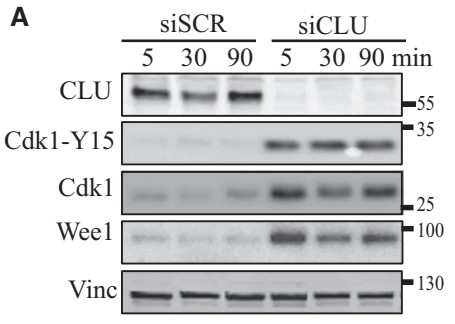

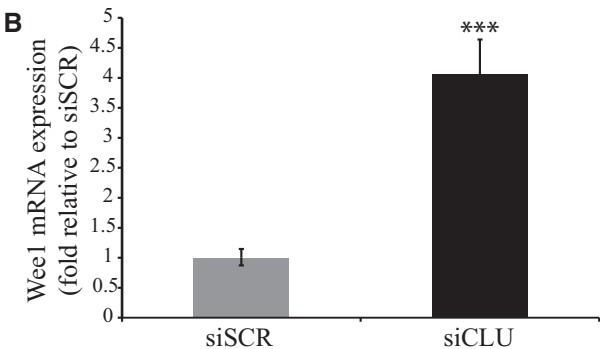

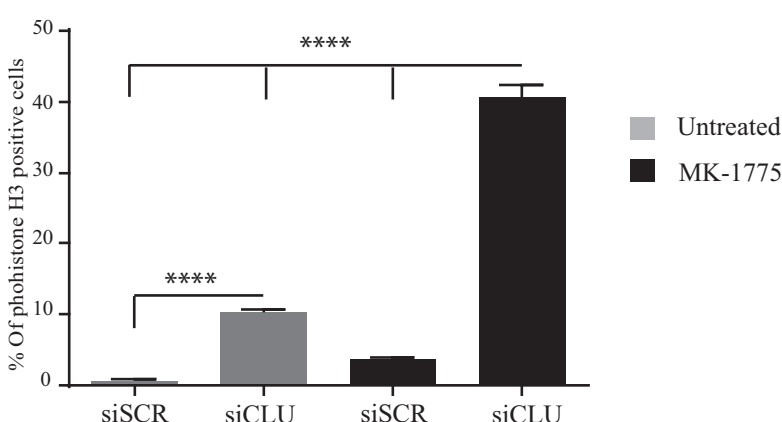

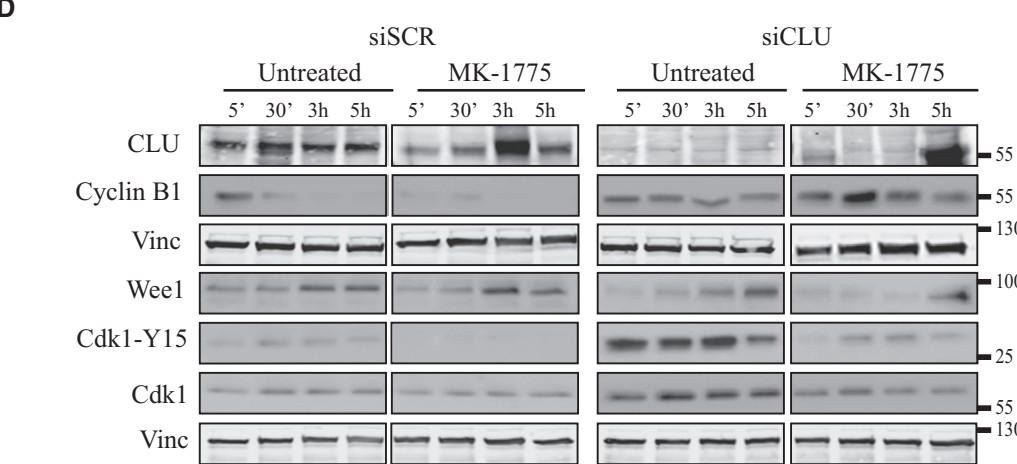

**Figure 7.  Up-regulation of Wee1 kinase compensates constitutive Cdk1 and Cdc25C activation in G2/M phase after CLU silencing.**

A    Western blot for CLU, Cdk1-Y15, Cdk1, and Wee1 in PC3 cells after siSCR or siCLU transfection followed by synchronization and nocodazole release at indicated time points. Vinculin was used as loading control.

B    Wee1 expression level was evaluated in PC3 cells after CLU silencing by real-time PCR. Error bar represents mean ± SEM, $n = 3$, ***$P < 0.001$ ($P = 0.0001$) by unpaired *t*-test followed by Welch's correction.

C    FACS analysis showing percentage of the PC3 cells positive for phosphohistone H3 after transfection with siSCR or siCLU in the presence and absence of Wee1 inhibitor, MK-1775. Error bar represents mean ± SEM, $n = 3$, ****$P < 0.0001$ by the Mann–Whitney test.

D    Western blot for CLU, Wee1, Cdk1-Y15, Cdk1, and cyclin B1 in PC3 cells after transfection with siSCR or siCLU followed by synchronization and treatment with or without MK-1775, a specific Wee1 inhibitor, for 12 h. Cells were harvested at different time points after nocodazole release. Vinculin was used as loading control.

Source data are available online for this figure.

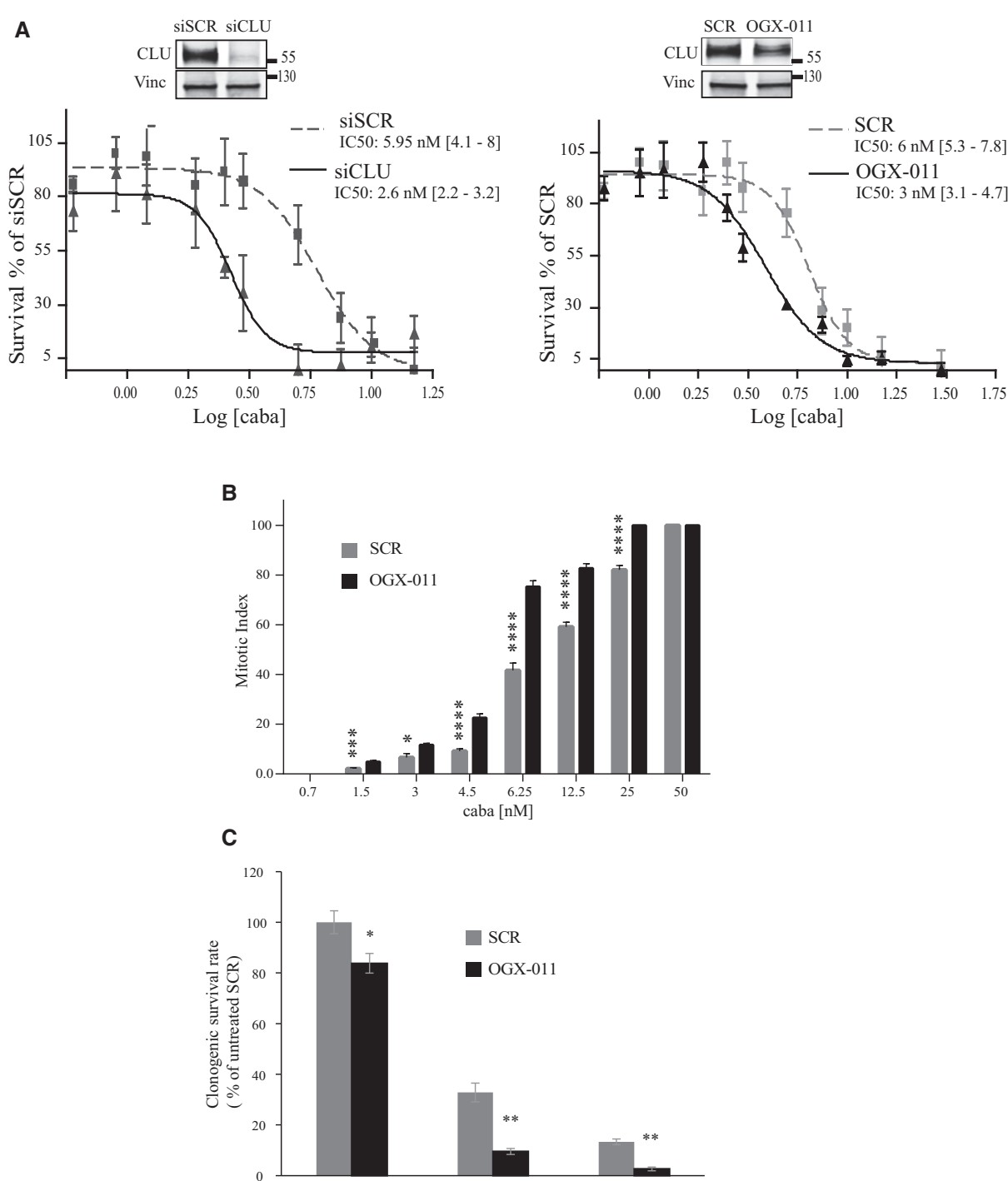

**Figure 8. CLU silencing increases cabazitaxel efficacy.**

A   Cabazitaxel IC50 calculation using WST1 assay in PC3 cells after transfection with siSCR or siCLU (left), and SCR or OGX-011 (right) followed by 72 h of cabazitaxel treatment. Percentage of surviving cells was calculated relative to control. Error bar represents mean $\pm$ SEM, $n$ = 3. Inset: Western blot was performed to verify CLU knockdown in cells.

B   Mitotic cells and cells in mitotic catastrophe were counted using DAPI in PC3 cells after SCR or OGX-011 transfection followed by different concentration of cabazitaxel as indicated (> 1,000 counts). Error bar represents mean $\pm$ SEM., *$P$ < 0.05, ***$P$ < 0.001, and ****$P$ < 0.0001 by Student's $t$-test (caba 1.5 nM $P$ = 0.00115; 3 nM $P$ = 0.0105; 4.5 nM $P$ = 1.93 $\times$ 10$^{-7}$; 6.25 nM $P$ = 8.9 $\times$ 10$^{-9}$; 12.5 nM $P$ = 6.8 $\times$ 10$^{-9}$; 25 nM $P$ = 1.2 $\times$ 10$^{-9}$).

C   Clonogenic assay in PC3 cells after SCR or OGX-011 transfection followed by treatment with different concentrations of cabazitaxel as indicated. The number of survival clones was assessed after 15 days using crystal violet assay. Error bars represent mean $\pm$ SEM $n$ = 3, *$P$ < 0.05, **$P$ < 0.01 by the Mann–Whitney test (0 nM caba $P$ = 0.039; 0.5 nM $P$ = 0.0079; 2 nM $P$ = 0.0079).

Figure 9.    CLU is required for rapid mitotic exit in cabazitaxel-resistant PC3 cells.

A    Cell cycle distribution of synchronized PC3 and PC3-R-caba cells 60 min after nocodazole release at indicated time points.
B    Western blot for CLU, Cdc25C, Cdc25C-T48, Cdk1, Cdk1-Y15, and Wee1 in IGR-CaP1 and PC3 parental (WT) or cabazitaxel-resistant (R-caba) cells as well as for PC3-R-caba cells transfected with siSCR or siCLU. Vinculin was used as loading control.
C    Cell survival measured using WST1 in PC3-R-caba cells after SCR or OGX-011 transfection and treatment with 10 nM cabazitaxel for 48 h. Error bar represents mean ± SEM, $n = 3$, ***$P < 0.001$ by one-way ANOVA followed by Bonferroni's *post hoc* correction.
D    Cell survival measured using WST1 in PC3 resistant cells after transfection with siSCR or siCLU followed by treatment with cabazitaxel (10 nM; C10), MK-1775 (10 nM; MK10), or both for 72 h. Error bar represents the mean ± SEM, $n = 3$, ***$P < 0.001$ by one-way ANOVA followed by Bonferroni's *post hoc* correction.
E    PARP cleavage was assessed by Western blot in cells treated as in (D) and its relative expression was quantified by densitometry using ImageJ. Vinculin was used as loading control.
F    Left panel: Quantification of tumor volume over 52 days in nude male mice xenotransplanted with PC3-docetaxel-resistant cells and treated with either MK-1775 (M, black line), MK-1775 and micellar taxane (MT, blue line), micellar taxane and OGX-011 (TO, red line), or MK-1775, micellar taxane and OGX-011 (MTO, green line). The error bars indicate mean ± SEM, $n = 8$. Right panel: Slopes of tumor growth between groups were compared by ANOVA followed by Dunnett's *post hoc* analysis. *$P < 0.05$, ***$P < 0.001$. $P = 0.0488$ (M to TO); $P = 0.0007$ (M to MTO).

confirming that co-targeting CLU and Wee1 overcomes cabazitaxel resistance. To confirm this hypothesis, PC3 cells resistant to docetaxel were inoculated into nude male mice, and when tumor average volume reached 150 mm$^3$, they were treated with MK-1775 alone (M), paclitaxel with MK-1775 (MT) or with OGX-011 (TO), and all three compounds together (MTO). As expected, combined docetaxel plus OGX-011 showed significant decreases in tumor volume compared to monotherapy ($P < 0.05$). However, the triple combination MTO has a more drastic effect ($P < 0.001$), confirming that CLU, Cdc25C and Wee1 are involved in cabazitaxel resistance. Collectively, these data suggest that co-targeting CLU and Wee1 can represent a novel approach to suppress adaptive survival responses and improve synergistic therapeutic responses in CRPC patients.

## Discussion

Resistance to chemotherapy remains a major challenge in the management of CRPC and other cancers. Recently, cabazitaxel, a semi-synthetic derivative of a natural taxoid, was shown to improve survival in docetaxel-recurrent mCRPC (de Bono *et al*, 2010). However, the benefit in overall survival is modest (median 2.4 months) at the cost of additional adverse effects. Therefore, it is imperative to understand taxane resistance mechanisms and develop new regimens that enhance therapeutic efficacy and reduce toxicity. The role of CLU in promoting emergence of a treatment-resistant phenotype has been previously established (Hoeller *et al*, 2005; Zoubeidi *et al*, 2010; Matsumoto *et al*, 2013; Niu *et al*, 2013; Xiu *et al*, 2013; Koltai, 2014), and recent studies from our group indicate that YB-1 transactivation of CLU in response to stress is a critical mediator of paclitaxel resistance (Shiota *et al*, 2011). In addition, several publications demonstrate how CLU down-regulation in combination with therapeutic stressors increases anti-cancer activity (Koltai, 2014). This synergy between CLU-KD and taxanes previously focused on the role of CLU in regulating stress-induced cell survival pathways involving Bax activation (Zhang *et al*, 2005) and autophagy (Zhang *et al*, 2014). However, it is well known that taxanes act on dividing cells and an important mechanism of cell death is mitotic catastrophe, suggesting that deregulated mitosis may mediate resistance.

In humans, *CLU* gene codes for two secretory isoforms (sCLU-1, sCLU-2) originating from transcriptional start sites in exons 1 and 2, respectively. sCLU is an ER-targeted, 449-aa polypeptide that represents the predominant translation product of the human gene (Cochrane *et al*, 2007; Rizzi & Bettuzzi, 2010). sCLU exerts its anti-apoptotic function as a molecular chaperone that stabilizes misfolded proteins under various stressors, retro-translocating from the endoplasmic reticulum under stress conditions to traffic to mito-chondria (Li *et al*, 2013) and autophagosomes (Zhang *et al*, 2014). In addition, previous reports described an alternative pro-apoptotic nuclear (nCLU) splice variant, lacking exon II and the ER signal peptide, that provokes G2/M arrest and mitotic catastrophe, associated with down-regulation of both cyclin B1 and Cdk1 when is overexpressed in PC3 cells (Scaltriti *et al*, 2004). Our results indicate that down-regulation of sCLU causes mitotic dysregulation leading to mitotic catastrophe. However, either sCLU silencing or overexpression of nCLU has different effects on MPF activity, possible due to lack of synchronization after overexpression of nCLU. Effects of CLU overexpression on cell cycle, resulting in G0/G1 accumulation, have been previously described in PNT2 immortalized human prostate epithelial cell line (Bettuzzi *et al*, 2002). To our knowledge, these are the only two studies that identify a possible role of CLU in cell cycle progression. In this study we provide novel insights into the previously unexplored role of CLU in mitosis in cancer cells showing that CLU affects mitosis exit by regulating Cdc25C activity. In particular, we demonstrate that CLU expression maintains low levels of activated Cdc25C and, moreover, that CLU and Cdc25C are negatively correlated *in vitro*, *in vivo*, and in human biopsies. The phosphatase Cdc25C is primarily involved in the mitotic exit through a complex set of kinase/phosphatase activity in response to various stimuli. PP2A has a major role in mitotic exit, and its failure to dephosphorylate Cdc25C at mitosis results in prolonged hyperphosphorylation and activation of Cdc25C, hence, delayed exit from mitosis (Forester *et al*, 2007). Consistent with this, our results indicate that CLU supports PP2A-mediated Cdc25C dephosphorylation and that CLU silencing decreases PP2A activity and delays mitotic exit through persistent Cdc25C and Cdk1 activation (Fig EV2B).

This novel function for CLU as a regulator of kinases and phosphatase activity in PP2A-mediated Cdc25C dephosphorylation is consistent with its role as a molecular chaperone that regulates folding and activation of many client proteins involved in signaling and transcriptional pathways. Ammar and Closset (2008) showed that CLU negatively regulated the PI3K–Akt pathway through attenuation of IGF-1. In another study Tang *et al* (2012) showed that knockdown of CLU by OGX-011 sensitized pancreatic cancer cells to gemcitabine by inhibition of

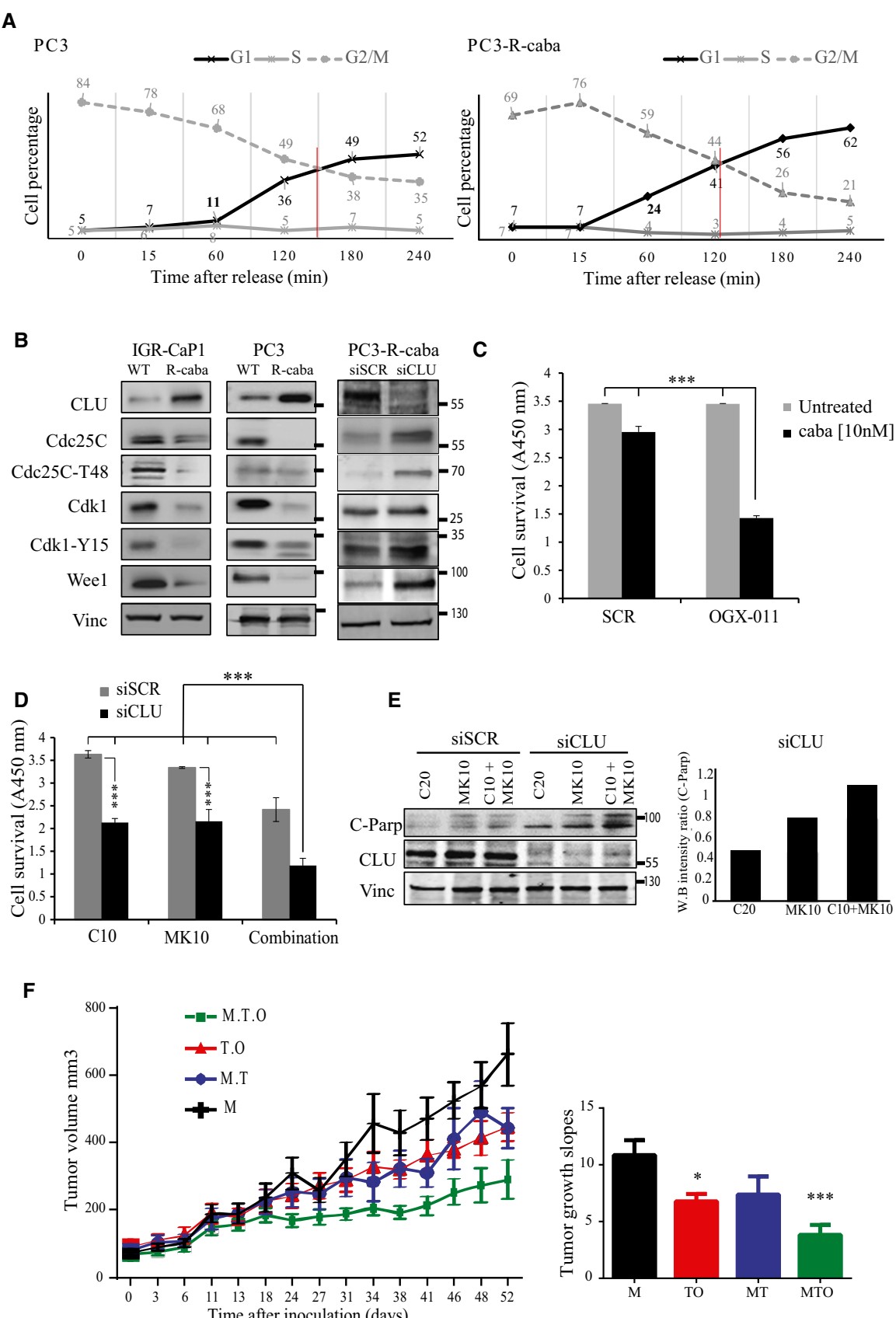

**Figure 9.**

gemcitabine-induced CLU-mediated pERK1/2 activation. We previously reported feed-forward links between stress-induced YB-1 to CLU activation in parallel with enhanced Akt and MAPK signaling which collectively support AR stability and activity during treatment with the AR antagonist enzalutamide (Matsumoto *et al*, 2013).

This study is the first to report an effect of CLU on a phosphatase involved in cell cycle. Mitosis is a complex mechanism controlled by the balance between many factors, such as Cdc25C and Wee1, which play antagonistic roles in mitosis progression (Perry & Kornbluth, 2007). Importantly, we show that unchecked Cdc25C activation, which occurs after CLU inhibition, can lead to cell death unless cells up-regulate protective mechanisms mediated through Wee1. Consistent with Cdc25C as a mitotic target of CLU, knock-down of CLU leads to compensatory up-regulation of Wee1 kinase to oppose the Cdc25C activity and permit cell survival. These mechanisms appear to be relevant *in vivo* in the context of combinatorial co-targeting regimens, as triple therapy targeting adaptive induction of CLU plus Wee1 optimally inhibited tumor growth compared to mono or double therapy regimens.

In summary, we present preclinical evidence demonstrating that CLU-KD significantly augments the anti-tumor activity of cabazitaxel in a mCRPC model. These preclinical data support the current phase III study of OGX-011 in combination with cabazitaxel as a second-line treatment of mCRPC (NCT01578655; clinicaltrials.gov). We also define for the first time that resistance to taxanes may occur via Cdc25C-Wee1-MPF regulation and that co-targeting Wee1 induction caused by CLU silencing represents a new strategy to address adaptive survival pathway activation and treatment resistance.

# Materials and Methods

### Prostate cancer cells and reagents

PC3 cells were purchased from American Type Culture Collection (ATCC) and authenticated with short tandem repeat (STR) profile analysis at Genetics Resources Core Facility at John Hopkins. IGR-CaP1 cells were provided by Dr Chauchereau (Gustave Roussy institute, France) (Chauchereau *et al*, 2011). IGR-CaP1 and PC3 resistant clones (IGR-CaP1-R-caba; PC3-R-caba) were obtained as previously described (Nakouzi *et al*, 2014). Briefly, resistant clones were selected by exposing cells to cabazitaxel in a dose-escalation manner. Initial culture was done with 1 nM cabazitaxel and cellular clones surviving were maintained in culture during at least 5 passages, followed by increasing cabazitaxel concentration up to 10 nM. PC3-docetaxel-resistant cells were established in our institution, as previously reported (Sowery *et al*, 2008). LNCaP cells were provided by Dr. Leland W.K. Chung (1992, MDACC, Houston, TX) and authenticated at GRCF. VCaP cells were purchased from ATCC and authenticated by ATCC's isoenzyme analysis. PC3 and VCAP cell lines were maintained in Dulbecco's modified Eagle's medium (DMEM; Invitrogen). LNCaP and IGR-CaP were maintained in RPMI 1640 (Invitrogen life Technologies, Inc) supplemented with 10% fetal bovine serum (FBS). All lines are tested for mycoplasma contamination regularly. Cabazitaxel was obtained from Sanofi-Aventis; nocodazole, crystal violet, okadaic acid, and thymidine

from Sigma-Aldrich; and WST1 from Roche. Cdc25C plasmid was purchased from Addgene (plasmid 10964).

### siRNA transfection and antisense

The CLU siRNA sequence corresponding to the initiation site in exon II of human CLU was 5′-GCAGCAGAGUCUUCAUCAU-3′, while the scrambled control siRNA sequence was 5′-CAGCGCUGACAACA-GUUUCAU-3′ (Dharmacon Research Inc., Lafayette, CO, USA). PP2A siRNA PP2A was purchased from Santa Cruz (sc-95565). OGX-011 and scrambled (SCR) oligonucleotide (ODN) with a 2′O-(2 methoxy) ethyl modification were supplied by OncoGenex Pharmaceuticals (Vancouver, BC). Cells were transfected using lipofectamine 2000 for siRNA (10 nM) and oligofectamine for ASO (100 nM) (Invitrogen Life Technologies Inc., Carlsbad, CA, USA) as previously described (Sowery *et al*, 2008).

### Cell viability and cell proliferation assays

Cells were transfected with CLU siRNA, OGX-011, or SCR control and 24 hrs later treated with cabazitaxel or vehicle. Cell viability in the treated plates was compared to untreated cells to calculate the surviving fraction using WST1 reagent. The dose–response curve and IC50 were estimated with a weighted 5-parameters logistic regression. Cell survival was evaluated using a standard colony forming assay (Nakouzi *et al*, 2014). One thousand cells/well were plated onto 6-well plates 48 h after transfection followed by the treatment with cabazitaxel. Fifteen days later, plates were stained with crystal violet and colonies were counted.

### Cell impedance assay (xCELLigence RTCA)

PC3 cells were seeded overnight 24 h after CLU silencing at 10,000 cells/well in 96-well E-plates (Roche). Measurements were taken continuously for ~60 h at 37°C. The RTCA software was used for data analysis, with statistical analysis using GraphPad Prism 5.0c (GraphPad Software Inc., San Diego, CA).

### Nocodazole release for mitotic progression

Cells were cultured in the appropriate complete media until 50–60% confluence. Media were removed and replaced with 2 mM thymidine for 18 h, and then, cells were washed three times with PBS and incubated with 100 ng/ml complete media plus nocodazole for an additional 12 h followed by three times washing with PBS. Cells were harvested at indicated time points by trypsinization, washed three times with cold PBS, and used for WB analysis or fixed for flow cytometry analysis.

### Flow cytometry analysis

Fixed synchronized cells were treated with 20 μg DNase-free RNase for 30 min and stained with propidium iodide (10 μg/ml) for 1 h. The percentage of cells in G1, S, G2, M, and sub-G1 phases was quantified using FACSCanto (Becton Dickinson) and analyzed with Cyflogic software. For phosphohistone H3 analysis, pellets were permeabilized in cold 0.1% Triton X-100, washed in FACS buffer (PBS, 2% FCS, 0.02% sodium azide, 0.5 mM EDTA), stained

with Alexa Fluor 647-conjugated phosphohistone H3 (Ser-10) antibody (no. 9716; Cell Signaling Technology) for 1 h at room temperature. Cells were washed and resuspended in PBS for FACS analysis.

### Real-time quantitative PCR assay

RNA extraction was done using TRIzol® RNA Isolation Reagents (Invitrogen Life Technologies, Inc.) according to manufacture instructions. PCR primers and probes for the following genes were purchased from Life Technology and used according to the manufacturer's recommendations (clusterin: Hs00156548_m1; Cdc25C: Hs00156411_m1; GAPDH: Hs02758991_g1; Wee1: Hs01119384_g1), using the ABI ViiA™ 7 Real-Time PCR detection system (Applied Biosystems) with a TaqMan Gene Expression Master Mix (Applied Biosystems). The amount of sample RNA was normalized by the amplification of glyceraldehyde-3-phosphate dehydrogenase (GAPDH) levels as internal control. The results are representative of at least 3 independent experiments.

### Western blot analysis

Whole-cell extracts were obtained by lysing the cells in an appropriate volume of ice-cold RIPA buffer composed of 50 mmol/l Tris–HCl, pH 7.4, 150 mmol/l NaCl, 0.5% sodium deoxycholate, 1% Nonidet P-40, 0.1% SDS containing 1 mmol/l $Na_3VO_4$, 1 mmol/l NaF, 1 mmol/l phenylmethylsulfonylfluoride, and protease inhibitor cocktail tablets (Complete; Roche Applied Science). Cellular extracts were clarified by centrifugation at $13,000 \times g$ for 10 min and protein concentrations of the extracts determined by a BCA protein assay kit (Thermo Scientific). Thirty micrograms of the extracts was boiled for 5 minutes in SDS sample buffer, separated by SDS–PAGE, and transferred onto a nitrocellulose membrane following standard methods. Membranes were probed with dilutions of primary antibodies Cdc25C (cat. Sc327), CLU (cat. Sc6419), and PP2A (cat. Sc107956) from Santa Cruz, p-Cdc25C-T48 (cat. 12028s), S216 (cat. 9528s), S198 (cat. 9529s), Cdk1 (cat. 77055s), p-Cdk1-Y15 (cat. 9111s), Wee1 (cat. 4936s), cyclin B1 (cat. 4135s), and pErk (cat. 9101s) from Cell Signaling, and vinculin (cat. V4505) from Sigma, followed by incubation with either horseradish peroxidase-conjugated secondary antibodies or fluorescent secondary antibodies. After washing, proteins were visualized by either a chemiluminescent detection system (GE Healthcare) or Odyssey Imaging System (LI-COR). Total proteins from LNCaP xenografts after CLU silencing were extracted using RIPA buffer and submitted to Western blot analysis as described above.

### Oligo microarray technology

Total RNA quality was assessed with the Agilent 2100 bioanalyzer prior to microarray analysis. Samples with a RIN value of $\geq 8.0$ were deemed acceptable for microarray analysis. Samples were prepared following Agilent's One-Color Microarray-Based Gene Expression Analysis Low Input Quick Amp Labeling v6.0. An input of 100 ng of total RNA was used to generate cyanine-3-labeled cRNA. Samples were hybridized on custom Agilent SurePrint G3 Human GE 4 × 180K Microarray (Design ID 032034). Arrays were scanned with the Agilent DNA Microarray Scanner at a 3 μm scan resolution, and data were processed with Agilent Feature Extraction 10.5.1.1.

Processed signal was quantile normalized with Agilent GeneSpring 11.5.1. To find significantly regulated genes between treatment groups, fold change (1.5) and *P*-values (< 0.05) gained from ANOVA (unequal variance) and unpaired *t*-tests were calculated. The *t*-tests were performed on normalized data that had been log-transformed, and the variances were not assumed to be equal. Multiple-testing correction using the Benjamini–Hochberg method was performed ($P \leq 0.05$). Data were analyzed through the use of QIAGEN's Ingenuity Pathway Analysis (IPA; QIAGEN, Redwood City) (www.quiagen.com/ingenuity). Statistically significant pathways involved in cellular functions were identified using the 2 × 2 contingency table Fisher's exact right-tailed test ($P < 0.05$; right-tailed Fisher's exact test).

### Kinexus phosphokinome analysis

Kinex™ KAM-880 Antibody Microarray was used to detect the changes in the expression levels and phosphorylation states after CLU silencing in PC3 prostate cells. Overall, 518 pan-specific antibodies (for protein expression) and 359 phosphosite-specific antibodies (for phosphorylation) listed on Kinexus Web site (http://www.kinexus.ca) were analyzed. A total of 50 μg of lysate protein from each sample was covalently labeled with a proprietary fluorescent dye combination. Free dye molecules were then removed at the completion of labeling reactions by gel filtration. After blocking and incubation the unbound proteins were washed away. The intensity of the signal on each spot corresponds to fluorescent captured proteins by the correspondent antibody for each sample. The images produced by each array were captured with a Perkin-Elmer ScanArray Reader laser array scanner (Waltham, MA). Signal quantification was performed with ImaGene 9.0 from BioDiscovery (El Segundo, CA). The background-corrected raw intensity data were logarithmically transformed with base 2. Since Z normalization in general displays greater stability as a result of examining where each signal falls in the overall distribution of values within a given sample, as opposed to adjusting all of the signals in a sample by a single common value, Z scores were calculated by subtracting the overall average intensity of all spots within a sample from the raw intensity for each spot, and dividing it by the standard deviations (SD) of all of the measured intensities within each sample. Z ratios were further calculated by taking the difference between the averages of the observed protein Z scores and dividing by the SD of all of the differences for that particular comparison. A Z ratio of ± 1.2 to 1.5 is inferred as significant (Cheadle *et al* 2003).

### Immunofluorescence microscopy

Prostate cancer cells were plated on coverslips in 12-well plates and transfected with siRNA for 24 h, followed by synchronization using thymidine/nocodazole blockage and released. Cells were fixed at different time points in PFA 4%, permeabilized with triton 0.5%, washed, blocked in PBS (3% BSA) and incubated with the anti-beta-tubulin (cat. 2146s; Cell Signaling) and anti-pericentrin (Ab-28144; Abcam) primary antibodies. Antigens were visualized using Alexa Fluor 488/594-conjugated antibodies. Nuclei were stained with DAPI Vectashield mounting reagent (Vector Laboratories). Images were acquired on LSM-780 confocal microscope at 63× magnification.

## *In situ* proximity ligation assay

Proximity ligation assay was performed to detect CLU and Cdc25C interaction using the CLU and Cdc25C antibodies. To visualize the bound antibody pairs, the Duolink Detection Kit (Duo92008) with PLA plus and minus probes for mouse and rabbit (Olink Bioscience) was used, according to the manufacturer's description. Cell slides were mounted with the Duolink Brightfield Mounting Medium (Olink Bioscience).

## PP2A assay

PP2A activity was assayed using the phosphatase kit V2460 (Promega, Madison, WI). Briefly, cells were lysed in phosphatase storage buffer, followed by removal of endogenous phosphate using the spin columns provided. Five-microgram protein samples in triplicate were incubated with phosphopeptide in PP2A-specific reaction buffer and RR(pt)VA peptide substrate for 30 min at room temperature. After washing, the samples were stained using the dye mixture for 15 min followed by OD measurement at 600 nm.

## Immunohistochemistry

This study was done on the total of 19 radical prostatectomy specimens obtained from Vancouver Prostate Centre Tissue Bank. The study protocol was approved by University of British Columbia (UBC) Clinical Research Ethics Board (CREB) and Vancouver Coast Hospital Research Institute Research (VCHRI) Research Ethics Board (REB). All patients gave informed consent as approved by UBC CREB and VCHRI REB. Immunohistochemical staining was conducted by Ventana autostainer model Discover XTTM (Ventana Medical System, Tuscan, Arizona) with enzyme-labeled biotin–streptavidin system and solvent-resistant DAB Map kit using 1/400 concentration of mouse monoclonal antibody against Cdc25C and 1/800 concentration of goat polyclonal antibody against CLU. All stained slides were digitalized with the SL801 autoloader and Leica SCN400 scanning system (Leica Microsystems, Concord, Ontario, Canada) at magnification equivalent to ×20 and subsequently stored in the SlidePath digital imaging hub (DIH; Leica Microsystems) of the Vancouver Prostate Centre. For each biomarker, representative cores (clearly positive, clearly negative, and mixed positive/negative) were manually identified by a pathologist and values on a four-point scale were assigned.

## *In vivo* studies

PC3-docetaxel-resistant cells ($2 \times 10^6$ cells) were injected subcutaneously into the right flank of $Foxn1^{nu}$ male NU/NU (NU-Foxn1nu) mice from ENVIGO (previously Harlan Laboratory) at the age of 5–6 weeks. Sample sizes were determined according to our previous *in vivo* assays (Zhang *et al*, 2014). Once tumors reached 150 mm³, mice were randomly divided into four groups: MK-1775 (M), MK-1775 and micellar taxane (MT), micellar taxane and OGX-011 (MO), and MK-1775–micellar taxane–OGX-011 (MTO), with eight mice in each group. Mice treated with MK-1775 (30 mg/kg, gavage) or OGX-011 (15 mg/kg, IP) received three consecutive treatments for 3 days before first cycle of taxane (day 18), then three times a week for 4 weeks. Mice treated with micellar taxane (15 mg/kg, IV) received the treatment three times a week for three cycles every

other week. Tumor growth was monitored using a caliper-measuring tool, and the three longest perpendicular axes in the $x/y/z$ plane of each tumor were measured. Investigators were blinded on grouping when measuring tumors. Tumor volume was calculated according to the standard formula: volume = $xy^2*0.5236$ (Janik *et al*, 1975). All animal procedures were carried according to the Canadian Council on Animal Care Guidelines and approved by the Institutional Animal Care Committee (IACC) at the University of British Columbia prior to conducting the study.

## Statistical analysis

The *in vitro* data were assessed using Student's *t*-test, ANCOVA, ANOVA and the Mann–Whitney test. The *t*-test was followed by Welch's correction when equal variance was not assumed. ANOVA was followed by a *post hoc* Bonferroni analysis. ANOVA followed by Dunnett's *post hoc* analysis was used to compare slopes of tumor growth for the *in vivo* experiment. GraphPad Prism software was used to calculate the statistical significance. The threshold of statistical significance was set at *$P < 0.05$, **$P < 0.01$, ***$P < 0.001$, and ****$P < 0.0001$. Exact *P*-values are indicated in the figure legends, when applicable.

**Expanded View** for this article is available online.

## Acknowledgements

This project was supported by a Terry Fox New Frontiers Program Project Grant TFF116129, a Prostate Cancer Foundation of British Columbia (PCFBC) Grant 2013, and a Prostate Cancer Canada Team Grant T2013-01. We would like to thank Mr. Dulguun Battsogt for his technical support and Mr. Robert Bell for his help with the statistical analysis.

## Author contributions

NAN, MG, and EB designed the research; NAN, CKW, EB, FZ, and LF performed the experiments; WJ, LN, AC, JB, SE, and YL provided useful reagents, data analysis, and helpful discussions; and MG and NAN wrote the manuscript.

---

**The paper explained**

**Problem**

Clusterin (CLU) is a stress-activated molecular chaperone that confers anti-cancer treatment resistance, and its inhibition potentiates the activity of taxanes in preclinical models. Despite this, treatment-resistant progression still occurs implicating additional compensatory survival mechanisms that may be exploitable once defined.

**Results**

Our data identify CLU as a key regulator of Cdc25C dephosphorylation and inactivation, through scaffolding interactions with the phosphatase PP2A, necessary for mitotic exit. CLU inhibition constitutively activates Cdc25C, delays mitotic exit and sensitizes cells to mitotic-targeting agents such as taxanes. However, constitutively activated Cdc25C leads to relief of negative feedback inhibition and adaptive induction of Wee1-Cdk1, to promote survival and limit therapeutic efficacy.

**Impact**

Co-targeting Wee1 induction with taxane/OGX-011 combinations represents a new strategy to address adaptive survival pathway activation and treatment resistance.

## Conflict of interest

By way of disclosure of conflict of interest, the University of British Columbia has submitted patent applications on OGX-011, listing Dr. Gleave as inventor. This IP has been licensed to OncoGenex Technologies, a Vancouver-based biotechnology company that Dr. Gleave has founding shares in.

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
