## [Review Process File · EMBO Molecular Medicine]

Clusterin knockdown sensitizes prostate cancer cells to taxane by modulating mitosis

Nader Al Nakouzi, Chris Kedong Wang, Eliana Beraldi, Wolfgang Jager, Susan Ettinger, Ladan Fazli, Lucia Nappi, Jenifer Bishop, Fan Zhang, Anne Chauchereau, Yohann Loriot and Martin Gleave

Corresponding author: Martin Gleave, University of British Columbia

Review timeline:

Submission date:	28 May 2015
Editorial Decision:	03 June 2015
Resubmission:	13 November 2015
Editorial Decision:	09 December 2015
Revision received:	17 March 2016
Editorial Decision:	08 April 2016
Revision received:	14 April 2016
Accepted:	18 April 2016

Transaction Report:

Editor: Roberto Buccione

1st Editorial Decision

03 June 2015

Thank you for the submission of your manuscript " Clusterin knockdown sensitizes prostate cancer cells to taxane by modulating mitosis" and many apologies for the taker longer than usual to get back to you. In fact, I wished to consult with an external expert, who was not immediately available.

I have now had the opportunity to read your manuscript and the related literature and I have also discussed it with my colleagues and the external advisor. I am afraid that we concluded that the manuscript does not appear well suited for publication in EMBO Molecular Medicine and have therefore decided not to proceed with peer review.

You show that clusterin knock-down increases sensitivity to cabazitaxel in a mCRPC model in vitro. Mechanistically, you suggest that resistance to taxanes may occur via Cdc25c-Wee1-MPF regulation. We realise that the data may support the current Phase III study of OGX-011 in combination with cabazitaxel as a second-line treatment of mCRPC and also appreciate that your findings suggest that co-targeting the Wee1 induction caused by CLU silencing may be a possible new strategy to address adaptive survival pathway activation and treatment resistance.

However, we agreed with the Advisor that the conceptual advance and broader impact provided by the work is somewhat limited by previous findings along the same line published by your group (albeit with less mechanistic insight) including on clusterin knockdown-mediated increase in

chemotherapy response in prostate cancer. We note that there has also been a report implicating CDC25 in docetaxel resistance. We do acknowledge the novelty arising from this manuscript describing the implication of a cell cycle phosphatase in CLU function.

Although I must return the manuscript to you at this stage, given our continued interest in your work we would be pleased to consider a new manuscript extending the study in a translational sense, e.g. offering proof of concept in an in vivo setting that indeed co-targeting CLU and Wee1 (or other CLU-affected cell cycle regulator) would provide potential benefit.

Please feel free to contact me if you wish to discuss further.

I am sorry I could not bring better news.

Resubmission

13 November 2015

Please find enclosed our manuscript entitled "Clusterin knockdown sensitizes prostate cancer cells to taxane by modulating mitosis", by Al-Nakouzi et. al, which I hope you will reconsider for publication in EMBO as an Original Article, after the addition of the in vivo results supporting its translational relevance, as suggested by the editor.

This manuscript is the first to report the role of Clusterin (CLU) on Cdc25C, a phosphatase involved in cell cycle. We show that CLU binds to Cdc25C and regulates its activity and that unchecked Cdc25c activation, which occurs after CLU inhibition, can lead to cell death unless cells upregulate protective mechanisms mediated through Wee1. Consistent with Cdc25C as a mitotic target of CLU, knockdown of CLU leads to compensatory transcriptional up-regulation of Wee1 kinase to oppose the Cdc25C activity and permit cell survival. We present pre-clinical evidence demonstrating that CLU-KD significantly augments the anti-tumor activity of cabazitaxel in a mCRPC model. These preclinical data support the current Phase III study of OGX-011 in combination with cabazitaxel as a second-line treatment of mCRPC (NCT01578655; clinicaltrials.gov). We also define, for the first time, that resistance to combinatorial CLU-KD plus taxanes may occur via Cdc25c-Wee1-MPF regulation, and that co-targeting Wee1 induction caused by CLU silencing, in vitro and in vivo, represents a new strategy to address adaptive survival pathway activation and treatment resistance.

We confirm that all authors have made a substantial contribution to the material submitted for publication, have read and approved the final manuscript and have no substantial direct or indirect commercial financial gain associated with the publication of the article. We also confirm that the manuscript or portions thereof are not

2nd Editorial Decision

09 December 2015

Thank you for the submission of your manuscript to EMBO Molecular Medicine. We are sorry that it has taken longer than usual to get back to you on your manuscript. In this case we experienced some difficulties in securing three appropriate expert reviewers and then obtaining their evaluations in a timely manner.

As you will see the three Reviewers are globally positive, but do raise many issues, some of which fundamental and mostly shared. Although I will not dwell into much detail, I would like to highlight the main points.

You will see that the one main recurring theme is the perceived inappropriateness of the statistical analysis. I should add that we fully agree that this is a current limitation of the manuscript and especially with respect to the specific points raised by reviewer 1. To this effect, please note that we are now requesting a complete author checklist to be submitted with all revised manuscripts (more on this below).

Reviewer 1 would also like you to validate the clusterin/Cdc25C interaction based on endogenous proteins and to verify whether Wee1 expression rescues the phenotype in clusterin-depleted cells. S/he also notes that most experimentation was performed on a single AR-cell line, which is considered a limitation. This reviewer also lists other items for your action.

Reviewer 2 is less reserved but does raise the issue of the precise role of PPA2 and the clusterin/Cdc25C interaction. S/he also mentions other points, including the need for careful editing and referencing.

Reviewer 3, in addition to his/her concerns on the statistical analysis and, again, the clusterin/Cdc25C interaction, the role of PP2A and the effects of Wee1 expression, points to other instances where the experiments are lacking sufficient controls, or insufficient data are presented to support specific claims. The points raised all have merits and require action.

In conclusion, while publication of the paper cannot be considered at this stage, given the potential interest of your findings, we have decided to give you the opportunity to address the above concerns. We are thus prepared to consider a substantially revised submission, with the understanding that the Reviewers' concerns must be addressed with additional experimentation as appropriate and that acceptance of the manuscript will entail a second round of review.

Please note that it is EMBO Molecular Medicine policy to allow a single round of revision only and that, therefore, acceptance or rejection of the manuscript will depend on the completeness of your responses included in the next, final version of the manuscript.

As you know, EMBO Molecular Medicine has a "scooping protection" policy, whereby similar findings that are published by others during review or revision are not a criterion for rejection. However, I do ask you to get in touch with us after three months if you have not completed your revision, to update us on the status. Please also contact us as soon as possible if similar work is published elsewhere.

As mentioned above, EMBO Molecular Medicine now requires a complete author checklist (<http://embomolmed.embopress.org/authorguide#editorial3>) to be submitted with all revised manuscripts. Provision of the author checklist is mandatory at revision stage; The checklist is designed to enhance and standardize reporting of key information in research papers and to support reanalysis and repetition of experiments by the community. The list covers key information for figure panels and captions and focuses on statistics, the reporting of reagents, animal models and human subject-derived data, as well as guidance to optimise data accessibility.

***** Reviewer's comments *****

Referee #1 (Comments on Novelty/Model System):

Most experiments were performed in only one prostate cancer cell line, which was androgen receptor negative.

Referee #1 (Remarks):

The authors identified an interesting role of Clusterin (CLU) in mitosis whereby prostate cancer cells depleted of CLU exhibited delayed exit from mitosis through the constitutive activation of

Cdc25C. Their data suggest that while CLU depletion sensitized prostate cancer cells to taxanes via regulation of Cdc25C and the phosphatase PP2A, compensatory up-regulation of Wee1 may contribute to acquired taxane resistance. The findings are potentially significant for prostate cancer therapy as the authors pursued approaches *in vivo* to bypass compensatory survival mechanisms following targeting of CLU. However, some conclusions are premature and the generality of the findings are uncertain as most of the experiments were conducted in a single cell line. Much of the statistical analysis is flawed or not adequately described.

Major points:

- 1) It is confusing that the authors identified Cdc25c through transcriptomics and phosphoproteomics conducted in different cell lines depleted of CLU.
- 2) Most experiments, with the exception of Fig 2, were done on a single cell line (PC3). Also, the rationale for using an androgen receptor negative cell line is unclear.
- 3) Is the regulation of Cdc25C mRNA/protein or Cdc25C activity more important mechanistically in cells depleted of CLU?
- 4) Figure 3 is confusing and requires clarification so as to not mislead the reader. How many mice were studied in Fig. 3A? Were differences significant? Are the data shown in 3B from xenografts of mice that were not treated with OGX-011? In Fig. 3C, how were the data normalized to Gleason score?
- 5) The immunoprecipitations in Fig 4A were done in cells overexpressing CLU and Cdc25C. Do endogenously expressed CLU and Cdc25C interact?
- 6) The authors conclude that Wee1 up-regulation is a compensatory survival response in CLU-depleted cells. An important experiment would be to determine whether overexpression of Wee 1 rescues the phenotype.
- 7) The authors should please describe the IGR-Cap-1 cells and their cabazitaxel derivative.
- 8) A section of "Statistical analysis" should be added in Materials and methods explaining software used and statistical approach of the data.
- 9) There are numerous issues related to the statistical analysis of the data:
 - a) Data should ideally be presented as mean +/- the standard error, instead of the mean +/- the standard deviation. The standard deviation is a measurement of data scatter around the mean (an index of dispersion), whereas the standard error measures accuracy of your estimation of the mean (Streiner, David L. "Maintaining standards: differences between the standard deviation and standard error, and when to use each." (1996): 498-502.)
 - b) All the Figure Legends should include the statistic value (for example: in a Student's T test the T value obtained, in an ANOVA analysis the F value, etc.).

In particular:

- Fig 1A: please specify statistical test/analysis used for the p value obtained
 Fig 1B: please perform a corresponding statistical analysis and report p values
 Fig 3C: please perform a corresponding statistical analysis and report p values
 Fig. 5C: the T test performed is inappropriate for the type of data being analyzed. ANOVA or its corresponding non parametric test (please see note below) should be conducted.
 Fig. 6D: please perform a corresponding statistical analysis and report p values
 Fig. 7 B: please specify statistical test/analysis used for the p value obtained
 Fig. 8C: the T test performed is inappropriate for the type of data being analyzed. The variances of the different groups do not seem to be equally distributed; therefore a T test cannot be performed without transforming the data or, alternatively, the use of a non-parametric test (please see note below)
 Fig. 9C, D and F: please perform a corresponding statistical analysis and report p values
 Note: Student's t-test should only be used when comparing two groups. For multiple group mean comparisons, the ANOVA test should be used. T test and ANOVA are only valid when the data are normally distributed.

Minor points:

- 1) In the Introduction: "Premature activation of Cdc25C is prevented by phosphorylation of specific residues in Cdc25C during interphase that are distinct from the sites phosphorylated in M phase" needs a reference. "the CLU inhibitor custirsen (OGX-011) failed to prolong survival when

combined with docetaxel in CRPC." needs a reference

2) In the Results section, in line 3 of the section "CLU knockdown modulates expression of mitosis regulation", the line "In PC3 cells, (...) and a 40% increase in population of G2/M cells (Fig. 1B)" should be replaced for " (...) and a 40% mean increase in population of G2/M cells (Fig. 1B)"

3) The phosphokinome analysis on Figure 1C, left panel, needs to be explained in Materials and Methods.

4) The p values for the non-parametric T test used for Figure 1C, left panel, should be included in the table or as supplemental information.

5) Reorganize Figure 2B so that the order of the cell lines matches the order of the data presented on Figure 2A

6) In Legends, on Figure 6 change (C) for (D) on line 8.

Referee #2 (Remarks):

This is an impressive study with important findings demonstrating a mechanism of action by which clusterin (CLU) inhibition stalls prostate cancer cell progression through mitosis, thus sensitizing them to taxane chemotherapeutics. This active area of research aims to enhance the efficacy of such therapy in the realm of prostate cancer, which has proven to be a challenging endeavor. The authors of this manuscript provide compelling evidence that CLU inhibition, which, alone, slows but does not eliminate prostate cancer cell proliferation, results in an accumulation of cells in G2/M phase and an up-regulation of cell cycle regulators Cdc25C and Cdc2. Further, the authors suggest that Cdc25C phosphorylation (T48)/activation, via loss of PP2A phosphatase activity, is an important means by which CLU knockdown inhibits mitotic exit. However, the simultaneous up-regulation of Wee1 after CLU inhibition, which acts in opposition of Cdc25C to maintain mitosis-promoting Cdc2 phosphorylation, represents a compensatory survival pathway that may underlie the failure of CLU inhibitors in clinical trials. Thus the authors conclude that taxane chemotherapeutic regimens may benefit from combinatorial treatment with CLU inhibitors, for sensitization, and Wee1 inhibitors, to overcome compensatory survival pathways.

In general, this study is very well done, and I believe the authors have thoroughly explored a complex pathway to reveal a very interesting mechanism that should be informative for future therapeutic development purposes. I have just a few comments that I hope the authors could address before this manuscript is published.

1) WST-1 assays should be validated by direct viable cell count.

2) The phosphatase assay described does not seem to be specific for PP2A, but instead measures total Ser/Thr phosphatase activity. The authors should clarify how their assay is specific, or if it is nonspecific, select another means of demonstrating that CLU knockdown disrupts PP2A activity.

3) Related to (2), the authors should more thoroughly establish that PP2A interaction with Cdc25C is disrupted by CLU knockdown.

4) Could the authors comment on why PP2A inhibition induces marked expression of CLU as well as Cdc25C and Cdc2? Induction of these cell cycle regulators along with CLU seems at odds with the authors' earlier observations.

5) Would PP2A inhibition and the resulting activation of Cdc25C be sufficient to overcome the Wee1/Cdc2 survival mechanism as the authors hint in their abstract? Fig 6C seems to suggest that PP2A silencing actually activates this survival mechanism similar to CLU inhibition.

6) Some figures are labeled/referenced incorrectly, and the manuscript should be carefully edited.

Referee #3 (Comments on Novelty/Model System):

The manuscript by Al Nakouzi et al. describes a large body of work that attempts to tie the biological consequences of decreasing clusterin expression in prostate cancer cells using siRNA and the specific biological effects that result. A major challenge to these studies is that the authors are attempting to both demonstrate the functional role of clusterin in mitosis (that remains poorly

understood) and the effects of decreasing its abundance on prostate cancer cell survival in response to taxols. The authors provide much data describing many pieces that together form a reasonably coherent story. However, much of the data requires better explanation, appropriate statistical analysis, further experimentation or proper controls.

Referee #3 (Remarks):

The remainder of this review will be organized to match the Results section of the manuscript, addressing each point the authors are trying to make.

1) CLU knockdown modulates expression of mitosis regulators

The authors state, "differential expression profiling identified many biologically-related gene clusters involved in the regulation of apoptosis, cell cycle progression and cell growth/proliferation." However, it appears that multiple t-tests were used to identify significant differences and there was no indication that any correction for multiple comparisons was performed. Therefore, it is unclear how many of the genes identified as significantly different between the two groups were significant only by chance. In Supp Fig 1A, the authors appear to have used something akin to gene set enrichment analysis (GSEA), although there is no description of such.

Fig 1A presents the effects of siCLU on a normalized cell index (derived from impedance measurements) that is a combined metric of cell number, cellular adhesion and cell-cell interactions; the authors should at least acknowledge the possibility that the change in impedance in response to siCLU could be due to changes in cellular effects other than proliferation. Even if the assumption is made that the majority of change in impedance is due to cell number changes and other effects are negligible, the rates of change for each curve are only different between 20 and 50 hours, suggesting that the effects of siCLU in the cell population are short-lived. Also, there appears to be a disconnect between the siRNA treatment effects on proteins and RNA levels of mitotic regulators that occur at 24 h after treatment and the effects on impedance, which appears to require another ~20 h before it diverges from control (assuming time 0 in Fig 1A is 24 h post siRNA addition.)

The authors state that they performed flow cytometry using pS10 histone H3 (pHH3) to identify cells in M phase as distinct from G2 phase but did not include these data in the graph shown in Fig 1B or describe the fraction of cells that are pHH3 positive. A delayed exit from mitosis would be expected to result in an increased fraction of cells staining positively for pHH3. This should be visible in an asynchronously dividing cell population.

No description was provided for how the Kinexus phosphokinome data were generated or interpreted.

The significance of the Venn diagram is not clear since the overlap between the analytes assessed by each of the two platforms is not described.

2) CLU silencing leads to increased levels and activation of Cdc25C in human cancer cells

Results support this in cell lines, although Fig 2A could be made stronger by making delta-ct the dependent variable or at least presenting the data in log scale. Inclusion of human PCa patient data adds strength to the conclusion; however, no explanation is provided as to why/how siCLU causes an increased transcription of cdc25C (which suggests an indirect rather than direct effect since clusterin is not known to regulate transcription). In addition, all immunoblot and IHC data could be quantified and presented in similar form to that shown on delta-ct plots (Fig 3B, right). Based on visual differences of the three siClu-treated tumors in Fig 3A it appears that there is a positive correlation between CLU levels and Cdc25C levels. Data presented in Fig 3C should also be quantified to add support for the authors' claim.

3) CLU binds to Cdc25C

This statement is not well supported by the data due to poor controls. A different protein should be used for IP and proximity ligation to control for nonspecific protein-protein interactions. Moreover, a single example of confocal colocalization is provided (Fig 4B) and the possibility that this interaction happened by chance was not considered. The majority of two colors do not appear to be colocalized within the cell.

4) CLU silencing leads to a delay in exit from mitosis

Flow cytometry assessing DNA content in nocodazole-blocked cells (Fig 5A) supports the authors' conclusion that reduced CLU delays mitotic exit; however, Fig 5C seems to indicate that nearly all cells are in mitosis 1 h after nocodazole release based on pHH3 levels. This is at odds with the data

presented in Fig 5A that indicates 50% of cells are in G1. These data could be combined and represented on a single 2-dimensional plot showing DNA content and pHH3 levels. Also, the effect of siCLU on the fraction of cells in mitosis (using pHH3 or some other indicator) should also be performed in asynchronously dividing cells (like data in Fig. 1).

No description is provided in the legend of Fig 5B about what arrows represent (segregation abnormalities?). These can/should be quantified from images to support the authors' claim.

Fig 5D provides reasonable evidence that reducing clusterin levels by siRNA results in delayed mitotic exit with increased levels of Cdc25C, Cdc2, and cyclin B1 after release of mitotic block with nocodazole.

5) CLU KD induces Cdc25C phospho-T48 accumulation through PP2A

Fig 6A does not show the level of total Cdc25C and does not have any positive control for the Cdc25C phosphosites. In addition, the specificity of the Cdc25C pT48 antibody used in Fig 6 has been called into question. Previous studies suggest that immunoreactive bands are detected by a Cdc25C pT48 antibody even when Cdc25C expression is reduced by siRNA and that this may be due to crossreactivity with PIP3K5, which has sequence identity with the Cdc25C pT48 epitope (doi:10.1371/journal.pone.0011798). Moreover, the authors do not show any evidence that CLU levels affect Cdc25C pT48 reactivity in the absence of nocodazole. This could be accomplished by immunofluorescence detection in mitotic cells.

6) Activation of Cdc25C by CLU-KD Is compensated by Wee1 up-regulation

The putative counterbalancing effects of siCLU and Wee1 should be verified in asynchronously dividing cells (not treated with nocodazole). This could be done by quantifying the fraction of cells treated with siCLU in mitosis by flow cytometry (or some other means) in the presence of the Wee1 inhibitor.

7) CLU silencing sensitizes PC3 cells to cabazitaxel

Interesting and important functional consequence of siCLU.

8) CLU-Cdc25C-Wee1 pathway is a resistance mechanism to cabazitaxel

These results are intriguing and appear to provide a rationale for the combining cabazitaxel with siCLU and a Wee1 inhibitor to enhance entry and delay exit from mitosis thereby increasing the likelihood that cells die during mitosis. However, much of the data lack appropriate controls. The immunoblots shown in Fig 9B appear to be on asynchronously dividing cells. The levels of the various cell cycle proteins may correspond to the fraction of cells in different phases of the cell cycle. Is there any evidence that the distributions of cell cycle positions are the same in the paired cell lines? Do the cell lines proliferate at the same rate? Also, Fig 9D-F do not have untreated controls (all samples have been treated with at least one drug).

Other questions and general comments:

How does siCLU result in increased CDC25C transcript levels? How does this fit with the authors' contention that CLU regulates mitotic entry by direct interaction with CDC25C?

The preferred name for Cdc2 is CDK1 (<http://www.ncbi.nlm.nih.gov/gene/?term=983>).

CLU-KD is never defined in the text. Clusterin knockdown? Cells treated with siCLU?

On page 7 the authors state, "A similar increase of Wee-1 was confirmed at mRNA level by quantitative PCR analysis (Fig. 7C)." However, Fig. 7C shows a Western blot of the effects of the Wee1 inhibitor MK-1775.

There are no size markers provided for any of the immunoblots.

It would be beneficial if the authors conferred with a statistician to determine which statistical tests are most appropriate and how to interpret them.

The colors of the bars in Fig 8C are not labeled and appear to be switched compared to the colors in Fig 8B.

Legend to Fig 9A does not match the figure.

Answers to reviewers:

Referee #1 (Remarks):

The authors identified an interesting role of Clusterin (CLU) in mitosis whereby prostate cancer cells depleted of CLU exhibited delayed exit from mitosis through the constitutive activation of Cdc25C. Their data suggest that while CLU depletion sensitized prostate cancer cells to taxanes via regulation of Cdc25C and the phosphatase PP2A, compensatory up-regulation of Wee1 may contribute to acquired taxane resistance. The findings are potentially significant for prostate cancer therapy as the authors pursued approaches in vivo to bypass compensatory survival mechanisms following targeting of CLU. However, some conclusions are premature and the generality of the findings are uncertain as most of the experiments were conducted in a single cell line. Much of the statistical analysis is flawed or not adequately described.

Major points:

- 1) It is confusing that the authors identified Cdc25c through transcriptomics and phosphoproteomics conducted in different cell lines depleted of CLU.

R1: We first collected transcriptome data in PC3 cells which indicated significant increases in Cdc25C mRNA levels after CLU silencing. To validate the functional role of Cdc25C in cells depleted of CLU we next analyzed protein and phosphor-protein levels in another prostate cancer cell line (LNCaP) to verify if the effect was more generalized or cell line specific.

We agree that the analysis from the same cell line would also have been appropriate; but we believe that similar results, confirmed in two different cell lines using two techniques, significantly strengthen our findings.

- 2) Most experiments, with the exception of Fig 2, were done on a single cell line (PC3). Also, the rationale for using an androgen receptor negative cell line is unclear.

R2: The reviewer raises the issue that most of the experiments were done in a single cell line, but several experiments were conducted using different cell lines, as specified below:

- *Figure 1C, the phosphokinome study was performed on LNCaP cells and then combined with transcriptome in PC3 to reduce the risk that pathways identified were cell line dependent.*
- *Figure 2A and 2B, VCaP and LNCaP cells were included in the study in addition to PC3.*
- *Figure 3A and 3B, LNCaP xenograft tumors were employed in this analysis.*
- *Figure 9B, we used IGRCaP-1 cell line for this study.*

After validating our observations in multiple cell lines, we decided to focus on one model, PC3 cells, to investigate the role of CLU on mitosis and taxane resistance. We agree that AR is the main driver of prostate cancer and that remains active in CRPC; however, the principal aim of our study was the analysis of the regulation of the cell cycle induced by CLU, independent of AR. In addition, they present the advantage to express higher endogenous CLU protein levels compared to LNCaP which make them a good model for CLU silencing experiments (see below). Moreover, PC3 cells represent a model of hyperproliferative AR-negative state (p53 null) with neuroendocrine features that can emerge after potent AR suppression. Therefore, PC3 cells represent a suitable model for our study.

- 3) Is the regulation of Cdc25C mRNA/protein or Cdc25C activity more important mechanistically in cells depleted of CLU?

R3: The regulation of Cdc25C is complex and largely modulated by phosphorylation. Phosphorylation regulates its stability, subcellular localization as well as catalytic activity. Cdc25C phosphorylation (and activation) is mainly controlled by the cdk1/cyclin B1 complex. At the end of mitosis, Cdc25C is dephosphorylated and degraded, facilitating exit from mitosis. Therefore, phosphatases are important regulators of cell cycle transition, and their protein levels and activities are tightly regulated. After CLU depletion, we observed an increase of both Cdc25C mRNA and

protein levels, as well as increased levels of the phosphorylated Cdc25C form which suggests that total protein levels increase but is also more active after CLU silencing. The increase of phosphorylation could be related to a decrease of a phosphatase (regulated by CLU) responsible of Cdc25C de-phosphorylation. Based on our findings, Cdc25C phosphorylation more likely precedes accumulation of protein even though both events are very important mechanistically.

- 4) Figure 3 is confusing and requires clarification so as to not mislead the reader. How many mice were studied in Fig. 3A? Were differences significant? Are the data shown in 3B from xenografts of mice that were not treated with OGX-011? In Fig. 3C, how were the data normalized to Gleason score?

R4: Figure 3A refers to an in vivo experiment where 6 mice were treated with SCR (as negative control) and 8 mice with OGX-011. In the figure, western blotting was performed in 3 representative animals for each group. Following the reviewer's request, the western blotting for Cdc25C and CLU proteins derived from all the mice included in the study is displayed in the figure below (left panel). The quantification of the protein levels, relative to the loading control, is reported as dot plots (right panel) and a significant difference was found between the two groups (t-test followed by Welch's correction. CLU $p=0.049$ and Cdc25C $p=0.024$). This new Fig has been replaced in the text. The legend has been up-dated.

- In Figure 3B, the mice were not treated with OGX-011 and this has been specified in the figure legend.

- The data included in Figure 3C derive from OGX-011 treated patients and they were normalized by clinical stage, Gleason score and serum PSA. For each patient, the pathologist selected the area with the highest Gleason score. This has been added in the figure legend.

- 5) The immunoprecipitations in Fig 4A were done in cells overexpressing CLU and Cdc25C. Do endogenously expressed CLU and Cdc25C interact?

R5: The interaction between CLU and Cdc25C occurs at a specific stage of mitosis, therefore detecting the binding between the two endogenous proteins, by immunoprecipitation, is technically challenging. However, following the reviewer's suggestion, we performed a co-ip in PC3 cells with endogenous CLU and Cdc25C after synchronization using thymidine/nocodazole block. The western blotting below shows an interaction between Cdc25C and CLU when cell lysate was immunoprecipitated with anti-CLU antibody followed by western blotting for Cdc25C or vice versa. Actin antibody was used as an additional negative control in one of the blot. In addition, the endogenously expressed CLU and Cdc25C interaction was supported by both immunofluorescence staining and proximity ligation assay (PLA) (Fig 4B and C in the text).

- 6) The authors conclude that Wee1 up-regulation is a compensatory survival response in CLU-depleted cells. An important experiment would be to determine whether overexpression of Wee 1 rescues the phenotype.

R6: Since Wee1 expression levels increase in response to CLU silencing, we believe that the proper way to confirm if Wee1 executes a compensatory response is actually to inhibit Wee1 induction, using MK-1775, a specific Wee 1 inhibitor. This experiment was already performed in Figs 9D-E and the results indeed suggest that inhibition of Wee1 induction triggered further cell death after silencing CLU. Moreover, the inhibition of Wee1, in absence of CLU, increases the mitotic population 4 times, as demonstrated by the increase of phosphohistone H3 determined by FACS analysis. This experiment was performed to answer to reviewer 3 question 17 and reported below.

- 7) The authors should please describe the IGR-Cap-1 cells and their cabazitaxel derivative.

R7: The description of the IGR-Cap-1 cell line is referred in the manuscript (Chauchereau et al 2011) and their cabazitaxel resistant derivative cells have been clarified in the materials and methods section.

- 8) A section of "Statistical analysis" should be added in Materials and methods explaining software used and statistical approach of the data.

R8: A section of statistical analysis has been added in the M&M of the manuscript and it's reported below.

Statistical Analysis

*The in vitro data were assessed using the Student t-test, ANCOVA, ANOVA and the Mann-Whitney test. T-test was followed by Welch correction when equal variance was not assumed. ANOVA was followed by a post-hoc Bonferroni analysis. ANOVA followed by Dunnett post-hoc analysis was used to compare slopes of tumor growth for the in vivo experiment. GraphPad Prism software was used to calculate the statistical significance. The threshold of statistical significance was set at * $p < 0.05$, ** $p < 0.01$, *** $p < 0.001$, **** $p < 0.0001$. Exact p values are indicated in the figure legends, when applicable.*

- 10) There are numerous issues related to the statistical analysis of the data: a) Data should ideally be presented as mean +/- the standard error, instead of the mean +/- the standard deviation. The standard deviation is a measurement of data scatter around the mean (an index of dispersion), whereas the standard error measures accuracy of your estimation of the mean (Streiner, David L. "Maintaining standards: differences between the standard deviation and standard error, and when to use each." (1996):498-502.)

R9: As requested by the reviewer the data were presented as mean +/- SEM, when applicable.

- 11) All the Figure Legends should include the statistic value (for example: in a Student's T test the T value obtained, in an ANOVA analysis the F value, etc.). In particular:

Q: Fig 1A: please specify statistical test/analysis used for the p value obtained

R: Data were analyzed using GraphPad PRISM. We performed a linear regression and an analysis of covariance (ANCOVA) test was used to demonstrate the statistical significance between the two groups. $p < 0.0001$. The statistical analysis used and p-value obtained were reported in the legend and in the M&M section.

Q: Fig 1B: please perform a corresponding statistical analysis and report p values

R: The experiment was performed three times and the cell percentage of the different populations in the two groups was compared using a paired t-test. G2/M population $p = 0.0058$; G1 $p = 0.008$; Fig1B was changed and the statistical analysis was added in the legend.

Q: Fig 3C: please perform a corresponding statistical analysis and report p values

R: As previously demonstrated CLU increases after NHT ($p = 0.0213$) and is significantly decreased after OGX-011 ($p = 0.012$). However, although there was a trend in Cdc25C increase after treatment with OGX-011, this did not achieve statistical significance, likely related to the low number ($n = 5$) of the samples that does not confer enough power to perform statistical analysis. This has been stated in the manuscript. Statistical analysis was performed with ANOVA followed by Bonferroni correction.

p value Clusterin - Untreated vs OGX treated	0.70
p value Clusterin - NHTtreated vs OGX treated	0.012
p value CDC25-Untreated vs OGX treated	0.717
p value CDC25- NHTtreated vs OGX treated	0.37
p value Clusterin - Untreated vs NHT	0.0213
p value CDC25C - Untreated vs NHT	0.160

Q: Fig. 5C: The T test performed is inappropriate for the type of data being analyzed. ANOVA or its corresponding non parametric test (please see note below) should be conducted.

R: Fig. 5 A and C have been replaced based on the request of the reviewer #3. The data were analyzed with ANOVA followed by Bonferroni's post hoc comparisons tests. The p values are reported in the Fig legend.

Q: Fig. 6D: please perform a corresponding statistical analysis and report p values

R: We performed one-way ANOVA followed by Bonferroni's post hoc comparisons tests. The p value was reported in the Fig legend.

Q: Fig. 7 B: please specify statistical test/analysis used for the p value obtained

R: We performed an unpaired T-test followed by a Welch's correction and the p value obtained was reported in the Fig legend.

Q: Fig. 8C: The T test performed is inappropriate for the type of data being analyzed. The variances of the different groups do not seem to be equally distributed; therefore, a t test cannot be performed without transforming the data or, alternatively, the use of a non-parametric test (please see note below)

R: According to the reviewer's suggestion, we performed a non-parametric test (Mann – Whitney test). The values obtained were reported in the Fig legend.

Q: Fig. 9C, D and F: please perform a corresponding statistical analysis and report p values Note: Student's t-test should only be used when comparing two groups. For multiple group mean comparisons, the ANOVA test should be used. T test and ANOVA are only valid when the data are normally distributed.

R: For Fig 9C and D we performed a one way ANOVA followed by Bonferroni's post hoc comparisons tests. The p values obtained were reported in the Fig legend.

For Fig 9F we performed a one way ANOVA followed by Dunnett's post hoc comparison test on the slopes. The p values obtained were reported in the Fig legend.

Minor points:

- 1) In the Introduction: "Premature activation of Cdc25C is prevented by phosphorylation of specific residues in Cdc25C during interphase that are distinct from the sites phosphorylated in M phase" needs a reference.

R1: References have been included.

- 2) "the CLU inhibitor custirsen (OGX-011) failed to prolong survival when combined with docetaxel in CRPC." needs a reference

R2: The reference has been.

- 3) In the Results section, in line 3 of the section "CLU knockdown modulates expression of mitosis regulation", the line "In PC3 cells, (...) and a 40% increase in population of G2/M cells (Fig. 1B)" should be replaced for "(...) and a 40% mean increase in population of G2/M cells (Fig. 1B)"

R3: This has been replaced

- 4) The phosphokinome analysis on Figure 1C, left panel, needs to be explained in Materials and Methods.

R4: The description of analysis on phosphokinome has been included in the Materials and Methods as below:

Kinexus phosphokinome analysis

Fifty μ g of lysate protein from each sample were covalently labelled with a proprietary fluorescent dye combination. Free dye molecules were then removed at the completion of labelling reactions by gel filtration. After blocking and incubation the unbound proteins were washed away. The images produced by each array were captured with a Perkin-Elmer ScanArray Reader laser array scanner (Waltham, MA). Signal quantification was performed with Imagen 9.0 from BioDiscovery (El Segundo, CA). The background-corrected raw intensity data were logarithmically transformed with base 2. Since Z normalization in general displays greater stability as a result of examining where each signal falls in the overall distribution of values within a given sample, as opposed to adjusting all of the signals in a sample by a single common value, Z scores were calculated by subtracting the overall average intensity of all spots within a sample from the raw intensity for each spot, and dividing it by the standard deviations (SD) of all of the measured intensities within each sample. Z ratios were further calculated by taking the difference between the averages of the observed protein Z scores and dividing by the SD of all of the differences for that particular comparison. A Z ratio of ± 1.2 to 1.5 is inferred as significant (Cheadle, Cho-Chung et al. 2003).

- 5) The p values for the non-parametric T test used for Figure 1C, left panel, should be included in the table or as supplemental information.

R5: All the genes listed in Figure 1C, left panel, had a p value ≤ 0.05 . This value has been specified in the legend of the figure.

- 6) Reorganize Figure 2B so that the order of the cell lines matches the order of the data presented on Figure 2A. In Legends, on Figure 6 change (C) for (D) on line 8.

R6: The Fig 2B has been reorganized and replaced in the text. Figure 6 legend has been corrected.

Referee #2 (Remarks):

This is an impressive study with important findings demonstrating a mechanism of action by which clusterin (CLU) inhibition stalls prostate cancer cell progression through mitosis, thus sensitizing them to taxane chemotherapeutics. This active area of research aims to enhance the efficacy of such therapy in the realm of prostate cancer, which has proven to be a challenging endeavor. The authors of this manuscript provide compelling evidence that CLU inhibition, which, alone, slows but does not eliminate prostate cancer cell proliferation, results in an accumulation of cells in G2/M phase and an up-regulation of cell cycle regulators Cdc25C and Cdc2. Further, the authors suggest that Cdc25C phosphorylation (T48)/activation, via loss of PP2A phosphatase activity, is an important means by which CLU knockdown inhibits mitotic exit. However, the simultaneous up-regulation of Wee1 after CLU inhibition, which acts in opposition of Cdc25C to maintain mitosis-promoting Cdc2 phosphorylation, represents a compensatory survival pathway that may underlie the failure of CLU inhibitors in clinical trials. Thus the authors conclude that taxane chemotherapeutic regimens may benefit from combinatorial treatment with CLU inhibitors, for sensitization, and Wee1 inhibitors, to overcome compensatory survival pathways.

In general, this study is very well done, and I believe the authors have thoroughly explored a complex pathway to reveal a very interesting mechanism that should be informative for future therapeutic development purposes. I have just a few comments that I hope the authors could address before this manuscript is published.

- 1) WST-1 assays should be validated by direct viable cell count.

R1: We believe that a validation of the WST-1 assay by a direct viable cell count is unnecessary since this test represents one of the most accurate and sensitive colorimetric assays to determine the number of viable eukaryotic cells. The test is based on the ability of viable metabolic active cells to reduce tetrazolium reagent and generate formazan products that are soluble in the culture media. The resulting signal is proportional to the number of viable cells [1, 2].

- 2) The phosphatase assay described does not seem to be specific for PP2A, but instead measures total Ser/Thr phosphatase activity. The authors should clarify how their assay is specific, or if it is nonspecific, select another means of demonstrating that CLU knockdown disrupts PP2A activity.

R2: The PP2A activity was determined with the Serine/Threonine Phosphatase Assay from Promega (Cat. #V2460). This kit contains RRA(pT)VA, a peptide substrate that is compatible with several serine/threonine phosphatases such as protein phosphatases 2A, 2B and 2C but is a poor substrate for protein phosphatase 1 because of its more stringent structural requirements. In particular, to detect PP2A activity, the company recommends the use of a PP2A specific reaction buffer that we used and specified in the material and methods section of the manuscript. Furthermore, the specificity of the reaction has been assessed in other publications as a reliable way to determine PP2A phosphatase activity.[3]

- 3) Related to (2), the authors should more thoroughly establish that PP2A interaction with Cdc25C is disrupted by CLU knockdown.

R3: It is well established that Cdc25 C phosphatase is a key regulator of Cdk1 activity and that PP2A negatively regulates Cdc25C during mitosis [4]. A prolonged hyperphosphorylation of Cdc25C, due to failure to its dephosphorylation by PP2A, is responsible of a constitutive activation of Cdk1 and delayed exit from mitosis. B56 is the only member of the B56 family of PP2A targeting subunit that binds Cdc25C and this binding controls a critical inhibitory site of Cdc25C [4]. Our results indicate accumulation of Cdc25C-T48 after CLU silencing due to a decreased PP2A activity; therefore it is reasonable to speculate that CLU could stabilize the complex between PP2A and Cdc25C, allowing PP2A to dephosphorylate Cdc25C, similar to its scaffolding roles in other PTMs

[5, 6]. In the absence of CLU this interaction could be attenuated leading to reduced Cdc25 dephosphorylation.

Following the reviewer's suggestion, we performed a proximity ligation assay (PLA) using Cdc25C and PP2A antibodies (left panel) in PC3 cells treated with siSCR and siCLU (see figure below). The interaction between Cdc25C and PP2A (red dots) was quantified using the Duolink image tool software. (right panel) indicating a significant decrease after CLU silencing. Actin was used as negative control. (**** $p < 0.0001$ by Kruskal-Wallis test followed by Dunn's multiple comparison test)

- 4) Could the authors comment on why PP2A inhibition induces marked expression of CLU as well as Cdc25C and Cdk2? Induction of these cell cycle regulators along with CLU seems at odds with the authors' earlier observations.

R4: Figure 6B shows that levels of CLU, Cdc25C and Cdk1 increase after okadaic acid treatment (PP2A inhibition). However, while increased Cdc25C and Cdk1 levels persist after knockdown of PP2A, the induction of CLU is not reproduced (Fig. 6C). Clusterin is a stress induced chaperone that is upregulated by cellular stress and therefore increased CLU levels are expected after OA (6).

- 5) Would PP2A inhibition and the resulting activation of Cdc25C be sufficient to overcome the Wee1/Cdk1 survival mechanism as the authors hint in their abstract? Fig 6C seems to suggest that PP2A silencing actually activates this survival mechanism similar to CLU inhibition.

R5: Our results indicate that CLU knockdown prolongs presence of T48 by decreasing PP2A activity. Since PP2A is downstream of CLU, as showed in Fig.6C, its down-regulation produces similar effect as CLU silencing, as pointed by the reviewer. However, its inhibition cannot overcome the Wee1/Cdk1 survival mechanism.

Our data suggest that an additional co-targeting strategy to overcome survival mechanisms is the combination of taxane with CLU knockdown and Wee1 inhibitor. To better clarify, a sentence in the abstract has been changed as follows:

“Simultaneous inhibition of CLU-regulated cell cycle effector wee1, may improve synergistic response of...”

- 6) Some figures are labeled/referenced incorrectly, and the manuscript should be carefully edited.

R6: The draft and the figures have been edited.

Referee #3 (Comments on Novelty/Model System):

The manuscript by Al Nakouzi et al. describes a large body of work that attempts to tie the biological consequences of decreasing clusterin expression in prostate cancer cells using siRNA and the specific biological effects that result. A major challenge to these studies is that the authors are attempting to both demonstrate the functional role of clusterin in mitosis (that remains poorly understood) and the effects of decreasing its abundance on prostate cancer cell survival in response to taxols. The authors provide much data describing many pieces that together form a reasonably coherent story. However, much of the data requires better explanation, appropriate statistical analysis, further experimentation or proper controls.

Referee #3 (Remarks):

The remainder of this review will be organized to match the Results section of the manuscript, addressing each point the authors are trying to make.

- 1) CLU knockdown modulates expression of mitosis regulators the authors state, "differential expression profiling identified many biologically-related gene clusters involved in the regulation of apoptosis, cell cycle progression and cell growth/proliferation." However, it appears that multiple t-tests were used to identify significant differences and there was no indication that any correction for multiple comparisons was performed. Therefore, it is unclear how many of the genes identified as significantly different between the two groups were significant only by chance.

R1: To find significantly regulated genes between treatment groups, fold change (1.5) and p-values (<0.05) gained from ANOVA (unequal variance) and unpaired t-tests were calculated. Multiple testing correction using Benjamini Hochberg was performed ($p \leq 0.05$). This sentence has been added to M&M section, Oligo Microarray Technology paragraph.

- 2) In Supp Fig 1A, the authors appear to have used something akin to gene set enrichment analysis (GSEA), although there is no description of such.

R2: Supp Fig.1 is now named Fig. EV1 as recommended by the journal guidelines. In Fig EV1.A 'data were analyzed through the use of QIAGEN's Ingenuity Pathway Analysis (IPA, QIAGEN, Redwood City, www.qiagen.com/ingenuity). ($P < 0.05$; right-tailed Fisher Exact Test). This is specified in the legend.

- 3) Fig 1A presents the effects of siCLU on a normalized cell index (derived from impedance measurements) that is a combined metric of cell number, cellular adhesion and cell-cell interactions; the authors should at least acknowledge the possibility that the change in impedance in response to siCLU could be due to changes in cellular effects other than proliferation. Even if the assumption is made that the majority of change in impedance is due to cell number changes and other effects are negligible, the rates of change for each curve are only different between 20 and 50 hours, suggesting that the effects of siCLU in the cell population are short-lived. Also, there appears to be a disconnect between the siRNA treatment effects on proteins and RNA levels of mitotic regulators that occur at 24 h after treatment and the effects on impedance, which appears to require another ~20 h before it diverges from control (assuming time 0 in Fig 1A is 24 h post siRNA addition).

R3: According to our previous data [7, 8], the effect of CLU downregulation on RNA and protein level is detectable at 24/48 hrs. post transfection, while the effect on cell proliferation requires additional time (a doubling time cycle). Moreover, as stated by the reviewer, the transfection was done 24 hours earlier than time 0 in a 10 cm plate and the same number of cells (10,000 per well)

were plated in the 96 well-plate for impedance measurements. Therefore, the effect of clusterin silencing on cell survival during the first 24 hours was not taken into account. As a result, the effect on cell survival was observed up to 72hr.

- 4) The authors state that they performed flow cytometry using pS10 histone H3 (pHH3) to identify cells in M phase as distinct from G2 phase but did not include these data in the graph shown in Fig 1B or describe the fraction of cells that are pHH3 positive. A delayed exit from mitosis would be expected to result in an increased fraction of cells staining positively for pHH3. This should be visible in an asynchronously dividing cell population.

R4: We confirmed the same results in asynchronous cells. The Fig below shows an increase of the percentage (from 0.8 to 10) of the cells positive for pHH3 after treatment with siCLU compared to siSCR. ($p < 0.0001$ Mann-Whitney).

As the reviewer suggested, the percentage of pHH3 positive cells in the asynchronous cells was added in the manuscript page 6 line 4 and the data was added in Fig.EV1.C

- 5) No description was provided for how the Kinexus phosphokinome data were generated or interpreted. The significance of the Venn diagram is not clear since the overlap between the analytes assessed by each of the two platforms is not described.

R5: A paragraph describing the kinexus was added in M&M and it is reported below.

Kinexus phosphokinome analysis

Fifty μ g of lysate protein from each sample were covalently labeled with a proprietary fluorescent dye combination. Free dye molecules were then removed at the completion of labeling reactions by gel filtration. After blocking and incubation the unbound proteins were washed away. The images produced by each array were captured with a Perkin-Elmer ScanArray Reader laser array scanner (Waltham, MA). Signal quantification was performed with ImaGene 9.0 from BioDiscovery (El Segundo, CA). The background-corrected raw intensity data were logarithmically transformed with base 2. Since Z normalization in general displays greater stability as a result of examining where each signal falls in the overall distribution of values within a given sample, as opposed to adjusting all of the signals in a sample by a single common value, Z scores were calculated by subtracting the overall average intensity of all spots within a sample from the raw intensity for each spot, and dividing it by the standard deviations (SD) of all of the measured intensities within each sample. Z ratios are further calculated by taking the difference between the averages of the observed protein Z scores and dividing by the SD of all of the differences for that particular comparison. A Z ratio of ± 1.2 to 1.5 is inferred as significant [61].

The Venn diagram was used to graphically illustrate the overlap of mRNA and proteins involved in the regulation of mitosis in both transcriptome and phosphokinome analysis. More details of the phosphokinome are explained in the paragraph describing the Kinexus antibody microarray. All genes listed in Figure 1C, left panel, had a p value ≤ 0.05 . This has been clarified in the figure legend.,

- 6) CLU silencing leads to increased levels and activation of Cdc25C in human cancer cells. Results support this in cell lines, although Fig 2A could be made stronger by making delta-ct the dependent variable or at least presenting the data in log scale.

R6: We believe that the quantitative qPCR data represented as fold change relative to siSCR in Fig.2A is a better way to visualize the correlation between CLU and Cdc25C levels. Similar representations of the data has been used in other publications from our group [8]

- 7) Inclusion of human PCa patient data adds strength to the conclusion; however, no explanation is provided as to why/how siCLU causes an increased transcription of cdc25C (which suggests an indirect rather than direct effect since clusterin is not known to regulate transcription).

R7: Understanding the mechanism underling the regulation of Cdc25C mRNA by CLU is very interesting but quite complex and would require dedicated extensive work. Our and other groups' publications demonstrated that CLU can regulate gene transcription through the modulation of some transcription factors (NF-kB, YB-1 and HSF-1 among others [5, 6, 9]). Therefore, CLU may regulate specific transcription factors (i. e. p53, NF-Y, E2F, Sp1/3) for Cdc25C and/ or some cell cycle-dependent element (CDE) in conjunction with a cell cycle gene homology region (CHR) [10-12].

- 8) In addition, all immunoblot and IHC data could be quantified and presented in similar form to that shown on delta-ct plots (Fig 3B, right). Based on visual differences of the three siClu-treated tumors in Fig 3A it appears that there is a positive correlation between CLU levels and Cdc25C levels. Data presented in Fig 3C should also be quantified to add support for the authors' claim.

R8: As suggested, Fig 3A has been replaced with a full set of samples and the corresponding quantification as dot plots, showing an inverse correlation, has been added in the manuscript. Statistical analysis using T test shows a significant difference. The bar-graph in Fig 3C (left panel) represents the IHC scores of the experiment presented in the right panel of the same figure.

- 9) CLU binds to Cdc25C. This statement is not well supported by the data due to poor controls. A different protein should be used for IP and proximity ligation to control for nonspecific protein-protein interactions. Moreover, a single example of confocal colocalization is provided (Fig 4B) and the possibility that this interaction happened by chance was not considered. The majority of two colors do not appear to be colocalized within the cell.

R9: We used IgG in the co-immunoprecipitation experiments which is an established negative control. However, following the reviewer's suggestion, we performed a co-IP with actin as an additional negative control (see figure below).

- For PLA, siCLU was used as negative control (Fig.4C), since the absence of the protein leads to an absence of interaction. However, actin and eIF4G were used as additional negative controls in a new PLA assay (see image below)

- Confocal microscopy assay has been double confirmed and a new image, (see below), where more interactions between CLU (red) and Cdc25C (green) were visible, has been replaced in Figure 4B.

- 10) CLU silencing leads to a delay in exit from mitosis. Flow cytometry assessing DNA content in nocodazole-blocked cells (Fig 5A) supports the authors' conclusion that reduced CLU delays mitotic exit; however, Fig 5C seems to indicate that nearly all cells are in mitosis 1 h after nocodazole release based on pHH3 levels. This is at odds with the data presented in Fig 5A that indicates 50% of cells are in G1. These data could be combined and represented on a single 2-dimensional plot showing DNA content and pHH3 levels.

*R10: We agree with the reviewer that there is a difference in the percentage of cells in mitosis between Fig.5A and 5C. This was due to the fact that the two experiments were performed independently. Therefore, following the reviewer's suggestion, we performed a co-staining with pHH3 and PI in PC3 cells. Figures below show a similar percentage of cells in mitosis and confirm a significant difference in both pHH3 positive staining and G2/M phase after CLU silencing (ANOVA followed by Bonferroni comparison; * $p=0.0142$, **** $p<0.0001$). We believe that the representation of these results with 2 separate histograms, rather than a single 2 dimensional plot, as suggested by the reviewer, best illustrates the effect of CLU silencing on pHH3 positive staining and G2/M. Therefore, we prefer to use this format in the manuscript. Accordingly, Fig 5A and 5C were replaced within the text and the statistical analysis added in the figure legend.*

- 11) Also, the effect of siCLU on the fraction of cells in mitosis (using pHH3 or some other indicator) should also be performed in asynchronously dividing cells (like data in Fig. 1).

R11: The asynchronous experiment is included in question 4 (see above). Moreover, as requested, this percentage has been specified in the text at page 6 line 4.

- 12) No description is provided in the legend of Fig 5B about what arrows represent (segregation abnormalities?). These can/should be quantified from images to support the authors' claim.

R12: The sentence "White arrows represent segregation abnormalities" was added in the legend of Fig. 5B.

Figure 5B was added to show that cells are blocked in mitosis after siCLU and to report that abnormalities were observed. Abnormalities are represented by more than 2 centrioles per cell, misalignment of chromosomes and malformation of the microtubule spindle. The quantification of these abnormalities is challenging and could not be quantified from images. Quantification has to be performed in 3D, it is not standardized and could be individual dependent. Taking this into account, we estimate that this percentage is 1-5 %.

- 13) Fig 5D provides reasonable evidence that reducing clusterin levels by siRNA results in delayed mitotic exit with increased levels of Cdc25C, Cdc2, and cyclin B1 after release of mitotic block with nocodazole.

- 14) CLU KD induces Cdc25C phospho-T48 accumulation through PP2A Fig 6A does not show the level of total Cdc25C and does not have any positive control for the Cdc25C phosphosites.

R14: Following the reviewer's suggestion, total Cdc25C western blot image has been added to Fig. 6A.

Increasing T48 levels after OA treatment represents the positive control for Cdc25C phosphosite, as shown in figure 6B.

- 15) In addition, the specificity of the Cdc25C pT48 antibody used in Fig 6 has been called into question. Previous studies suggest that immunoreactive bands are detected by a Cdc25C pT48 antibody even when Cdc25C expression is reduced by siRNA and that this may be due to crossreactivity with PIP3K5, which has sequence identity with the Cdc25C pT48 epitope (doi:10.1371/journal.pone.0011798).

R15: The questionable pT48 antibody mentioned by the reviewer (from the referred paper) is a polyclonal antibody purchased from Cell Signalling Technologies [13]. However, the antibody used in our experiment is a monoclonal antibody (cat # 12028).

In addition, the specificity of our antibody is clearly confirmed in Fig 6C where downregulation of PP2A, known to dephosphorylate the T48, induces an increase of this phosphosite.

- 16) Moreover, the authors do not show any evidence that CLU levels affect Cdc25C pT48 reactivity in the absence of nocodazole. This could be accomplished by immunofluorescence detection in mitotic cells.

R16: The effect of siCLU on T48 is shown in Fig. 6B

In addition, following the reviewer's suggestion, we performed T48 immunofluorescence staining in absence of nocodazole after siSCR and siCLU transfection. The image below clearly shows increased T48 after siCLU treatment.

- 17) Activation of Cdc25C by CLU-KD is compensated by Wee1 up-regulation the putative counterbalancing effects of siCLU and Wee1 should be verified in asynchronously dividing cells (not treated with nocodazole). This could be done by quantifying the fraction of cells treated with siCLU in mitosis by flow cytometry (or some other means) in the presence of the Wee1 inhibitor.

R17: Following the reviewer's suggestion, we determined the percentage of positive pHH3 stained cells using a flow cytometry analysis in asynchronized PC3 cells treated with siSCR/siCLU in

presence and absence of Wee1 inhibitor, MK-1775. Fig. below (the same as the reply to question 6 of the review #1) shows that inhibition of Wee1, in absence of CLU, further increases the mitotic population, delaying the mitosis exit and increasing sensitivity to taxane.

- 18) CLU silencing sensitizes PC3 cells to cabazitaxel Interesting and important functional consequence of siCLU. CLU-Cdc25C-Wee1 pathway is a resistance mechanism to cabazitaxel these results are intriguing and appear to provide a rationale for the combining cabazitaxel with siCLU and a Wee1 inhibitor to enhance entry and delay exit from mitosis thereby increasing the likelihood that cells die during mitosis. However, much of the data lack appropriate controls. The immunoblots shown in Fig 9B appear to be on asynchronously dividing cells. The levels of the various cell cycle proteins may correspond to the fraction of cells in different phases of the cell cycle. Is there any evidence that the distributions of cell cycle positions are the same in the paired cell lines? Do the cell lines proliferate at the same rate?

R18: As expected and described, the parental and the resistant derivative cells do not proliferate at the same rate [14, 15]. However, the cell cycle distribution of the parental PC3 and their derivative resistant clones is identical as shown in figure below. This suggests that the difference in cell cycle protein levels is not a consequence of the cell cycle distribution.

- 19) Also, Fig 9D-F do not have untreated controls (all samples have been treated with at least one drug). Do the cells have to grow in the presence of the drug to maintain the resistance phenotype?

R19: The resistant cells used in this experiment are constantly maintained in the presence of drugs. Therefore, a non-treated condition is not necessary.

- 20) Other questions and general comments: How does siCLU result in increased CDC25C transcript levels? How does this fit with the authors' contention that CLU regulates mitotic entry by direct interaction with CDC25C?

R21: As stated in response 7, the effect of CLU on Cdc25C transcription regulation has not been established. While we cannot exclude a direct or indirect effect on the transcription, we demonstrated that CLU regulates Cdc25C protein expression through a direct interaction of the two proteins (CLU and Cdc25C).

- 21) The preferred name for Cdc2 is CDK1 (<http://www.ncbi.nlm.nih.gov/gene/?term=983>).

R22: We substituted Cdc2 with Cdk1

- 22) CLU-KD is never defined in the text. Clusterin knockdown? Cells treated with siCLU?

R23: CLU-KD means CLU knock down, and siCLU means siRNA targeting CLU mRNA. These are now stated in the revised version of the manuscript, page 6 line 17 and page 4 line 7, respectively.

- 23) On page 7 the authors state, "A similar increase of Wee-1 was confirmed at mRNA level by quantitative PCR analysis (Fig.7C)." However, Fig. 7C shows a Western blot of the effects of the Wee1 inhibitor MK-1775.

R23: This has been corrected in the revised version.

- 24) There are no size markers provided for any of the immunoblots.

R24: Molecular weight markers have been included to all the immunoblots in the revised version.

- 25) It would be beneficial if the authors conferred with a statistician to determine which statistical tests are most appropriate and how to interpret them.

R25: The statistical analysis information has been included in a new paragraph (Statistical analysis) in the Material and Methods section as reported below.

*The in vitro data were assessed using the Student t-test, ANCOVA, ANOVA and the Mann-Whitney test. T-test was followed by Welch correction when equal variance was not assumed. ANOVA was followed by a post-hoc Bonferroni analysis. ANOVA followed by Dunnett post-hoc analysis was used to compare slopes of tumor growth for the in vivo experiment. GraphPad Prism software was used to calculate the statistical significance. The threshold of statistical significance was set at * $p < 0.05$, ** $p < 0.01$, *** $p < 0.001$, **** $p < 0.0001$. Exact P-values are indicated in the figure legends, when applicable.*

- 26) The colors of the bars in Fig 8C are not labeled and appear to be switched compared to the colors in Fig 8B.

R26: This has been corrected in the revised version.

- 27) Legend to Fig 9A does not match the figure.

R27: The figure legend has been corrected.

References:

1. Terry L Riss, P., Richard A Moravec, BS, Andrew L Niles, MS, Helene A Benink, PhD, Tracy J Worzella, MS, and Lisa Minor, *Cell Viability Assays*. Assay Guidance Manual, 2015.
2. Stoddart, M.J., *WST-8 Analysis of Cell Viability During Osteogenesis of Human Mesenchymal Stem Cells*. Mammalian Cell Viability: Methods and Protocols, Methods in Molecular Biology. **740**.
3. Bae, D. and S. Ceryak, *Raf-independent, PP2A-dependent MEK activation in response to ERK silencing*. Biochem Biophys Res Commun, 2009. **385**(4): p. 523-7.
4. Forester, C.M., et al., *Control of mitotic exit by PP2A regulation of Cdc25C and Cdk1*. Proc Natl Acad Sci U S A, 2007. **104**(50): p. 19867-72.
5. Zoubeidi, A., et al., *Clusterin facilitates COMMD1 and I-kappaB degradation to enhance NF-kappaB activity in prostate cancer cells*. Mol Cancer Res, 2010. **8**(1): p. 119-30.
6. Zhang, F., et al., *Clusterin facilitates stress-induced lipidation of LC3 and autophagosome biogenesis to enhance cancer cell survival*. Nat Commun, 2014. **5**: p. 5775.
7. Lejl-Garolla, B., et al., *Hsp27 Inhibition with OGX-427 Sensitizes Non-Small Cell Lung Cancer Cells to Erlotinib and Chemotherapy*. Mol Cancer Ther, 2015. **14**(5): p. 1107-16.
8. Yamamoto, Y., et al., *Generation 2.5 antisense oligonucleotides targeting the androgen receptor and its splice variants suppress enzalutamide-resistant prostate cancer cell growth*. Clin Cancer Res, 2015. **21**(7): p. 1675-87.
9. Shiota, M., et al., *Clusterin Is a Critical Downstream Mediator of Stress-Induced YB-1 Transactivation in Prostate Cancer*. Molecular Cancer Research, 2011. **9**(12): p. 1755-1766.
10. Haugwitz, U., et al., *A single cell cycle genes homology region (CHR) controls cell cycle-dependent transcription of the cdc25C phosphatase gene and is able to cooperate with E2F or Sp1/3 sites*. Nucleic Acids Res, 2002. **30**(9): p. 1967-76.
11. St Clair, S., et al., *DNA damage-induced downregulation of Cdc25C is mediated by p53 via two independent mechanisms: One involves direct binding to the cdc25C promoter*. Molecular Cell, 2004. **16**(5): p. 725-736.
12. Manni, I., et al., *NF-Y mediates the transcriptional inhibition of the cyclin B1, cyclin B2, and cdc25C promoters upon induced G(2) arrest*. Journal of Biological Chemistry, 2001. **276**(8): p. 5570-5576.
13. Wang, R., et al., *Regulation of Cdc25C by ERK-MAP kinases during the G2/M transition*. Cell, 2007. **128**(6): p. 1119-32.
14. Corcoran, C., et al., *Docetaxel-resistance in prostate cancer: evaluating associated phenotypic changes and potential for resistance transfer via exosomes*. PLoS One, 2012. **7**(12): p. e50999.
15. Nakouzi, N.A., et al., *Targeting CDC25C, PLK1 and CHEK1 to overcome Docetaxel resistance induced by loss of LZTS1 in prostate cancer*. Oncotarget, 2014. **5**(3): p. 667-78.

3rd Editorial Decision

08 April 2016

Thank you for the submission of your revised manuscript to EMBO Molecular Medicine. We have now received the enclosed reports from the referees that were asked to re-assess it.

As you will see, although the reviewers are now globally supportive, Reviewer 3 has a few remaining concerns that I would ask you to deal with. S/he is mainly concerned that you did not present much of the data generated during the revision process and still maintains that some datasets remain poorly described.

After further discussion with my colleagues, we agreed that inclusion of the additional data and the clarifications would be ultimately be useful for the readership and would do justice to your work. I would therefore ask you to carefully consider all the points raised and the inclusion of additional data in the main figures where appropriate, or an Appendix or additional Expanded View figures. You will not be required to perform any additional experimentation at this point.

I will proceed with an editorial decision on your next, final version. Please upload and additional copy of your manuscript with the changes highlighted (e.g. in red lettering) when re-submitting your

revision.

Please submit your revised manuscript as soon as possible.

I look forward to reading a final revised version of your manuscript as soon as possible.

***** Reviewer's comments *****

Referee #1 (Remarks):

The authors have addressed my comments adequately.

Referee #2 (Remarks):

No further comments

Referee #3 (Remarks):

General comments

The manuscript by Al Nakouzi and others describes a novel role of clusterin in the regulation of mitosis and provides evidence that decreasing the expression of clusterin by siRNA alters mitotic progression to enhance sensitivity to taxanes. Moreover, the authors provide evidence that compensatory upregulation of Wee1 when clusterin expression is inhibited by siRNA is an actionable target for the treatment of taxane-resistant prostate cancer.

The authors are providing a substantial amount of data to support their claims, including newly generated data in response to the reviewers' requests, but the information is still not presented in a clear and concise way. It is unclear why much of the newly generated data is only provided in the response to reviewers and not included as Supplementary information. In addition, some of the experimental data described in the main text remain poorly described, making it difficult to assess their validity and detracting from the overall importance of their observations. However, most of these instances are peripheral to the authors' primary observation that decreasing clusterin expression in prostate cancer cells lengthens their time spent in mitosis and sensitizes them to taxane treatment. The complexity of mitosis regulation poses a significant challenge to presenting a coherent story about how clusterin may be involved and how its dysregulation can affect response of prostate cancer cells to taxane treatment. Intriguingly, the authors provide a cartoon model of the effects of decreased clusterin expression on the molecular regulators of mitosis duration and the different response to taxanes in presence and absence of clusterin that summarizes their work fairly well, including the importance of the various molecular observations; yet this reviewer found no reference to it anywhere in the text, including the response to reviewers. Overall, this manuscript provides important, novel observations of a role for clusterin in regulation of mitosis and the potential of using this knowledge for augmenting response of prostate cancer treated with taxanes. However, the manuscript could be significantly improved by more clearly describing the how the specific molecular observations in response to decreasing clusterin expression would be expected to affect mitotic progression.

Specific claims made by the authors (in order of presentation in manuscript)

CLU knockdown modulates expression of mitosis regulators

This is primarily supported by the demonstration of PCa cells treated with siCLU have increased immunoblot reactivity of proteins known to regulate mitosis entry and exit, and is further supported by many subsequent immunoblots throughout the manuscript. In addition, the increased fraction of siCLU-treated cells with 4N DNA content (considered in G2 or M phase of the cell cycle) supports a functional role for CLU in the regulation of mitosis. However, the inadequate description of the "phosphokinome" data detracts from these findings, since their interpretation is tied to the transcriptome data. Although the authors provided some details about the "phosphokinome" data, it

was insufficient to interpret the importance of the overlap with the transcriptome data (see Specific responses to reviewers' concerns below). The transcriptome and "phosphokinome" data do little to support the authors' claim.

CLU silencing leads to increased levels and activation of Cdc25C in human cancer cells

This is relatively well supported in Figures 2 and 3, including a revision of Figure 3A that shows increased Cdc25C in LNCaP xenografts treated with siCLU. However, the authors did not consider the request to plot the expression levels of Cdc25C and CLU simultaneously for each tumor as in the mRNA levels shown in Figure 3B. The previous request for quantification of data in Figure 3C was erroneous and should have referred to the immunoblot data shown in Figure 3B.

CLU binds to Cdc25C

New immunoprecipitation data support the direct interaction between endogenous Cdc25C and CLU, although the possibility that CLU nonspecifically copurified with Cdc25C is not ruled out by the inclusion of IgG alone as a control. Specificity of the interaction is best demonstrated by precipitation of a different protein from the lysate without copurifying CLU. The addition of a new Figure 4B better supports the colocalization of Cdc25C and CLU using confocal microscopy. Curiously, the authors provide good controls for the proximity ligation assay in their response to reviewers but do not include these data in the main figure or add them as Supplementary Information. The addition of the negative controls significantly increases the strength of the claim and should be included.

CLU silencing leads to a delay in exit from mitosis

Addition of new flow cytometry data showing increased pHH3 levels in asynchronously dividing cells treated with siCLU or an inhibitor of Wee1 supports their roles in lengthening time spent in mitosis. Other data provided also support this sufficiently.

CLU KD induces Cdc25C phospho-T48 accumulation reducing PP2A phosphatase activity

Figure 6A now includes a blot showing total Cdc25C levels, although the bands do not appear to match the bands detected by the phospho-specific antibodies. In addition, the authors provide immunofluorescence data demonstrating the increased levels of pT48 Cdc25C in response to siCLU in their response to reviewers. Again, it is unclear why the authors would not include these data or even refer to them in the main text.

Activation of Cdc25C by CLU-KD is compensated by Wee1 up-regulation

Upon suggestion from this reviewer, new data are provided that show combination of siCLU and a Wee1 inhibitor leads to a substantial increase in the fraction of pHH3-positive cells in asynchronously dividing cells. The fact that this strong result has been left out of the main text (or even Supplementary Information) is puzzling. Nevertheless, the authors' have sufficiently demonstrated that Wee1 expression is increased in response to siCLU.

CLU silencing sensitizes PC3 cells to cabazitaxel

Data are sufficient to support this interesting observation.

CLU-Cdc25C-Wee1 pathway is a resistance mechanism to cabazitaxel

Authors have provided data in response to reviewers that demonstrates the paired sensitive and taxane-resistant cell lines have highly similar cell cycle distributions suggesting that the differences in cell cycle regulated proteins are not merely reflections of different abundance of cells in specific cell cycle positions. Again, the authors have not considered that this information would be useful to include in the main text.

Specific responses to reviewers' concerns

Reviewer#3.1 and 2: It is unclear how many of the genes identified as significantly different between the two groups were significant only by chance. The authors appear to have used something akin to gene set enrichment analysis (GSEA), although there is no description of such. No description was provided for how the Kinexus phosphokinome data were generated or interpreted.

The authors addressed these concerns by providing the statistical tests for inclusion of the genes in

the list. This was a necessary addition. However, these statements were also meant to question the comparison between the transcript levels and the "phosphokinome" values. The authors have provided technical details of sample preparation for the "phosphokinome" values, but have not fully described what is actually being measured. What exactly is the "signal" that is being measured? I.e., how were proteins actually detected? How many and which proteins were actually quantified apart from the ones shown? The Kinexus website (<http://www.kinexusproducts.ca>) appears to have numerous products available. Which one was used? This is basic information for which there is no excuse not to include.

Reviewer#3.5: The significance of the Venn diagram is not clear since the overlap between the analytes assessed by each of the two platforms is not described.

Authors' response R5: "The Venn diagram was used to graphically illustrate the overlap of mRNA and proteins involved in the regulation of mitosis in both transcriptome and phosphokinome analysis. More details of the phosphokinome are explained in the paragraph describing the Kinexus antibody microarray. All genes listed in Figure 1C, left panel, had a p value {less than or equal to}0.05. This has been clarified in the figure legend."

Again, this fails to address the main question about the overlap between the analytes measured in the two approaches.

Reviewer#3.1: The authors should at least acknowledge the possibility that the change in impedance in response to siCLU could be due to changes in cellular effects other than proliferation.

The authors did not respond to this comment at all.

Reviewer#3.2: In Supp Fig 1A, the authors appear to have used something akin to gene set enrichment analysis (GSEA), although there is no description of such.

Authors' response R2: Supp Fig.1 is now named Fig. EV1 as recommended by the journal guidelines

In Fig EV1.A 'data were analyzed through the use of QIAGEN's Ingenuity Pathway Analysis (IPA, QIAGEN, Redwood City, www.qiagen.com/ingenuity). (P<0.05; right-tailed Fisher Exact Test). This is specified in the legend.

This does not adequately describe how the data shown in Fig EV1A were produced or how they should be interpreted.

Reviewer#3.6: CLU silencing leads to increased levels and activation of Cdc25C in human cancer cells Results support this in cell lines, although Fig 2A could be made stronger by making delta-ct the dependent variable or at least presenting the data in log scale.

Authors' response R6: We believe that the quantitative qPCR data represented as fold change relative to siSCR in Fig.2A is a better way to visualize the correlation between CLU and Cdc25C levels. Similar representations of the data has been used in other publications from our group [8]

The suggestion for a change to log2 scale was to better show both increases and decreases from control on the same scale. For example, a value of 0.25 (the approximate level of CLU in siCLU-treated VCaP) represents a 4-fold decrease.

Reviewer#3.8: In addition, all immunoblot and IHC data could be quantified and presented in similar form to that shown on delta-ct plots (Fig 3B, right). Based on visual differences of the three siClu-treated tumors in Fig 3A it appears that there is a positive correlation between CLU levels and Cdc25C levels. Data presented in Fig 3C should also be quantified to add support for the authors' claim.

Authors' response R8: As suggested, Fig 3A has been replaced with a full set of samples and the corresponding quantification as dot plots, showing an inverse correlation, has been added in the manuscript. Statistical analysis using T test shows a significant difference. The bar-graph in Fig 3C (left panel) represents the IHC scores of the experiment presented in the right panel of the same

figure.

While the new graphs support the authors' conclusions that the siCLU-treated xenografts have higher Cdc25C, the authors did not produce the suggested graph that compares the levels of Cdc25C and CLU simultaneously in the same samples, which is directly obtainable from the immunoblot data. The reference to Figure 3C was inadvertent and was meant to refer to the immunoblots in Figure 3B, which has not been quantified.

Reviewer#3.15: [Question regarding the specificity of the Cdc25 pT48 antibody]

The specific catalog numbers for the antibodies have now been provided, alleviating this concern.

Reviewer#3.19: Also, Fig 9D-F do not have untreated controls (all samples have been treated with at least one drug). Do the cells have to grow in the presence of the drug to maintain the resistance phenotype?

Authors' response R19: The resistant cells used in this experiment are constantly maintained in the presence of drugs. Therefore, a non-treated condition is not necessary.

According to the legend, the data presented in Figure 9D and E were from "PC3 cells after transfection with siSCR or siCLU followed by treatment with cabazitaxel," not resistant cell lines.

Other issues

The correlation coefficient value in the legend to Fig EV.1B: "In silico correlation analysis of Cdc25C and CLU mRNA levels in 460 prostate cancer patients using GeneSapiens data set. (Spearman correlation: rho -0.86; p-value = 0.0107)" do not match the values described in the text on page 4, "(p<0.001; r = -0.23)."

A description of the GeneSapiens resource has not been provided. A simple reference to the URL of the resource would be very useful.

Spelling error on page 13: Fifteen days later

2nd Revision - authors' response

14 April 2016

Reviewer report, 14 April 2016

General comments

The manuscript by Al Nakouzi and others describes a novel role of clusterin in the regulation of mitosis and provides evidence that decreasing the expression of clusterin by siRNA alters mitotic progression to enhance sensitivity to taxanes. Moreover, the authors provide evidence that compensatory upregulation of Wee1 when clusterin expression is inhibited by siRNA is an actionable target for the treatment of taxane-resistant prostate cancer.

The authors are providing a substantial amount of data to support their claims, including newly generated data in response to the reviewers' requests, but the information is still not presented in a clear and concise way. It is unclear why much of the newly generated data is only provided in the response to reviewers and not included as Supplementary information. In addition, some of the experimental data described in the main text remain poorly described, making it difficult to assess their validity and detracting from the overall importance of their observations. However, most of these instances are peripheral to the authors' primary observation that decreasing clusterin expression in prostate cancer cells lengthens their time spent in mitosis and sensitizes them to taxane treatment. The complexity of mitosis regulation poses a significant challenge to presenting a

coherent story about how clusterin may be involved and how its dysregulation can affect response of prostate cancer cells to taxane treatment. Intriguingly, the authors provide a cartoon model of the effects of decreased clusterin expression on the molecular regulators of mitosis duration and the different response to taxanes in presence and absence of clusterin that summarizes their work fairly well, including the importance of the various molecular observations; yet this reviewer found no reference to it anywhere in the text, including the response to reviewers. Overall, this manuscript provides important, novel observations of a role for clusterin in regulation of mitosis and the potential of using this knowledge for augmenting response of prostate cancer treated with taxanes. However, the manuscript could be significantly improved by more clearly describing the how the specific molecular observations in response to decreasing clusterin expression would be expected to affect mitotic progression.

The cartoon model the reviewer is referring to is not part of the manuscript. The journal asked the authors to provide a synthetic image that summarizes the main finding of the discovery only after the manuscript has been accepted for publication.

Specific claims made by the authors (in order of presentation in manuscript) CLU knockdown modulates expression of mitosis regulators This is primarily supported by the demonstration of PCa cells treated with siCLU have increased immunoblot reactivity of proteins known to regulate mitosis entry and exit, and is further supported by many subsequent immunoblots throughout the manuscript. In addition, the increased fraction of siCLU-treated cells with 4N DNA content (considered in G2 or M phase of the cell cycle) supports a functional role for CLU in the regulation of mitosis. However, the inadequate description of the "phosphokinome" data detracts from these findings, since their interpretation is tied to the transcriptome data. Although the authors provided some details about the "phosphokinome" data, it was insufficient to interpret the importance of the overlap with the transcriptome data (see Specific responses to reviewers' concerns below). The transcriptome and "phosphokinome" data do little to support the authors' claim.

1. The microarray and phosphokinome data were used at the beginning of the project as screening to identify CLU-modulated pathways. The fact that both techniques convergently detected an increase of Cdc25C and other mitosis regulator genes after CLU silencing, strengthened the importance of clusterin in modulating mitosis in prostate cancer models and provided a rationale to continue our study. Further clarification of the phosphokinome antibody based array indicating the name of the assay, the number of proteins detected and the method of the detection were added in the M&M section of the manuscript.

>CLU silencing leads to increased levels and activation of Cdc25C in human cancer cells This is relatively well supported in Figures 2 and 3, including a revision of Figure 3A that shows increased Cdc25C in LNCaP xenografts treated with siCLU. However, the authors did not consider the request to plot the expression levels of Cdc25C and CLU simultaneously for each tumor as in the mRNA levels shown in Figure 3B. The previous request for quantification of data in Figure 3C was erroneous and should have referred to the immunoblot data shown in Figure 3B.

2. The graphic representation suggested by the reviewer is indeed an interesting way to plot the data, however is not suitable to represent the results since we have two different conditions (SCR and OGX011). Therefore, we prefer to leave the previous quantification graph as the most appropriate method to analyze the data and apply adequate statistical analysis test.

The protein expression of Cdc25C and CLU in Figure 3B has been quantified and added in the manuscript.

CLU binds to Cdc25C

>New immunoprecipitation data support the direct interaction between endogenous Cdc25C and CLU, although the possibility that CLU nonspecifically copurified with Cdc25C is not ruled out by the inclusion of IgG alone as a control. Specificity of the interaction is best demonstrated by precipitation of a different protein from the lysate without copurifying CLU. The addition of a new Figure 4B better supports the colocalization of Cdc25C and CLU using confocal microscopy. Curiously, the authors provide good controls for the proximity ligation assay in their response to reviewers but do not include these data in the main figure or add them as Supplementary Information. The addition of the negative controls significantly increases the strength of the claim and should be included.

3. New immunoprecipitation and additional PLA negative controls have been added in supplementary data Fig. EV.1 C and D

> CLU silencing leads to a delay in exit from mitosis Addition of new flow cytometry data showing increased pHH3 levels in asynchronously dividing cells treated with siCLU or an inhibitor of Wee1 supports their roles in lengthening time spent in mitosis. Other data provided also support this sufficiently.

> CLU KD induces Cdc25C phospho-T48 accumulation reducing PP2A phosphatase activity Figure 6A now includes a blot showing total Cdc25C levels, although the bands do not appear to match the bands detected by the phospho-specific antibodies. In addition, the authors provide immunofluorescence data demonstrating the increased levels of pT48 Cdc25C in response to siCLU in their response to reviewers. Again, it is unclear why the authors would not include these data or even refer to them in the main text.

4. Immunofluorescent data were added as figure 6A right panel in the manuscript.

Activation of Cdc25C by CLU-KD is compensated by Wee1 up-regulation Upon suggestion from this reviewer, new data are provided that show combination of siCLU and a Wee1 inhibitor leads to a substantial increase in the fraction of pHH3-positive cells in asynchronously dividing cells. The fact that this strong result has been left out of the main text (or even Supplementary Information) is puzzling. Nevertheless, the authors have sufficiently demonstrated that Wee1 expression is increased in response to siCLU.

5. The pHH3 data have been included in the manuscript as Fig. 7 C.

CLU silencing sensitizes PC3 cells to cabazitaxel Data are sufficient to support this interesting observation. CLU-Cdc25C-Wee1 pathway is a resistance mechanism to cabazitaxel Authors have provided data in response to reviewers that demonstrates the paired sensitive and taxane-resistant cell lines have highly suggesting that the differences in cell cycle regulated proteins are not merely reflections of different abundance of cells in specific cell cycle positions. Again, the authors have not considered that this information would be useful to include in the main text.

6. The data of cell cycle distributions of the taxane-resistant cell lines have been added in Fig. EV.2A

Specific responses to reviewers' concerns

>Reviewer#3.1 and 2: It is unclear how many of the genes identified as significantly different between the two groups were significant only by chance. The authors appear to have used something akin to gene set enrichment analysis (GSEA), although there is no description of such. No description was provided for how the Kinexus phosphokinome data were generated or interpreted. The authors addressed these concerns by providing the statistical tests for inclusion of the genes in the list. This was a necessary addition. However, these statements were also meant to question the comparison between the transcript levels and the "phosphokinome" values. The authors have provided technical details of sample preparation for the "phosphokinome" values, but have not fully described what is actually being measured. What exactly is the "signal" that is being measured? I.e., how were proteins actually detected? How many and which proteins were actually quantified apart from the ones shown? The Kinexus website (<http://www.kinexusproducts.ca>) appears to have numerous products available. Which one was used? This is basic information for which there is no excuse not to include.

The requested information has been added into the M&M section of the manuscript as highlighted in red below.

7. Kinex™ KAM-880 Antibody Microarray was used to detect the changes in the expression levels and phosphorylation states after CLU silencing in PC3 prostate cells. Overall, 518 pan-specific antibodies (for protein expression) and 359 phosphosite-specific antibodies (for phosphorylation) listed on kinexus website (<http://www.kinexus.ca>) were analyzed. Fifty µg of lysate protein from each sample were covalently labelled with a proprietary fluorescent dye combination. Free dye molecules were then removed at the completion of labelling reactions by gel filtration. After blocking and incubation the unbound proteins were washed away. The intensity of the signal on each spot corresponds to fluorescent captured proteins by the correspondent antibody for each sample. The images produced by each array were captured with a Perkin-Elmer ScanArray Reader laser array scanner (Waltham, MA). Signal quantification was performed with ImaGene 9.0 from BioDiscovery (El Segundo, CA). The background-corrected raw intensity data were logarithmically transformed with base 2. Since Z normalization in general displays greater stability as a result of

examining where each signal falls in the overall distribution of values within a given sample, as opposed to adjusting all of the signals in a sample by a single common value, Z scores were calculated by subtracting the overall average intensity of all spots within a sample from the raw intensity for each spot, and dividing it by the standard deviations (SD) of all of the measured intensities within each sample. Z ratios were further calculated by taking the difference between the averages of the observed protein Z scores and dividing by the SD of all of the differences for that particular comparison. A Z ratio of ± 1.2 to 1.5 is inferred as significant (Cheadle, Cho-Chung et al. 2003).

>Reviewer#3.5: The significance of the Venn diagram is not clear since the overlap between the analytes assessed by each of the two platforms is not described.

>Authors' response R5: "The Venn diagram was used to graphically illustrate the overlap of mRNA and proteins involved in the regulation of mitosis in both transcriptome and phosphokinome analysis. More details of the phosphokinome are explained in the paragraph describing the Kinexus antibody microarray. All genes listed in Figure 1C, left panel, had a p value {less than or equal to}0.05. This has been clarified in the figure legend."

Again, this fails to address the main question about the overlap between the analytes measured in the two approaches.

8. Similar issue was raised by this reviewer. We added a more detailed description of the phosphokinome analysis that should clarify his concerns in points 1 and 7. Briefly, we identified all the genes involved in mitosis regulation in both transcriptome and phosphokinome microarrays. Of all these genes, only those significantly modulated after clu silencing were listed in figure 1C and those overlapping were represented in the Venn diagram.

>Reviewer#3.1: The authors should at least acknowledge the possibility that the change in impedance in response to siCLU could be due to changes in cellular effects other than proliferation.

>The authors did not respond to this comment at all.

9. We acknowledge the possibility that a change in impedance in response to siCLU could be due to changes in cellular effects other than proliferation. However, as the reviewer implied in his first review and widely accepted (PMID:21765947), we assumed that the majority of change in impedance is due to cell number changes and other effects are negligible.

>Reviewer#3.2: In Supp Fig 1A, the authors appear to have used something akin to gene set enrichment analysis (GSEA), although there is no description of such.

>Authors' response R2: Supp Fig.1 is now named Fig. EV1 as recommended by the journal guidelines In Fig EV1.A 'data were analyzed through the use of QIAGEN's Ingenuity Pathway Analysis (IPA, QIAGEN, Redwood City, www.qiagen.com/ingenuity).(P<0.05; right-tailed Fisher Exact Test). This is specified in the legend.

This does not adequately describe how the data shown in Fig EV1A were produced or how they should be interpreted.

10. More details regarding how the IPA data were produced was added in the M&M section of the manuscript

Reviewer#3.6: CLU silencing leads to increased levels and activation of Cdc25C in human cancer cells Results support this in cell lines, although Fig 2A could be made stronger by making delta-ct the dependent variable or at least presenting the data in log scale.

> Authors' response R6: We believe that the quantitative qPCR data represented as fold change relative to siSCR in Fig.2A is a better way to visualize the correlation between CLU and Cdc25C levels. Similar representations of the data has been used in other publications from our group [8]

> The suggestion for a change to log2 scale was to better show both increases and decreases from control on the same scale. For example, a value of 0.25 (the approximate level of CLU in siCLU-treated VCaP) represents a 4-fold decrease.

11. We agree with the reviewer that both methods (fold change or ΔCT) can be used to show the decrease or increase of mRNA levels. However, being equal, we prefer to represent the data in fold changes as we have previously reported.

>Reviewer#3.8: In addition, all immunoblot and IHC data could be quantified and presented in similar form to that shown on delta-ct plots (Fig 3B, right). Based on visual differences of the three siClu-treated tumors in Fig 3A it appears that there is a positive correlation between CLU levels and Cdc25C levels. Data presented in Fig 3C should also be quantified to add support for the authors' claim.

>Authors' response R8: As suggested, Fig 3A has been replaced with a full set of samples and the corresponding quantification as dot plots, showing an inverse correlation, has been added in the manuscript. Statistical analysis using T test shows a significant difference. The bar-graph in Fig 3C (left panel) represents the IHC scores of the experiment presented in the right panel of the same figure.

While the new graphs support the authors' conclusions that the siCLU-treated xenografts have higher Cdc25C, the authors did not produce the suggested graph that compares the levels of Cdc25C and CLU simultaneously in the same samples, which is directly obtainable from the immunoblot data. The reference to Figure 3C was inadvertent and was meant to refer to the immunoblots in Figure 3B, which has not been quantified.

12. This question has been already answered in point 2.

>Reviewer#3.15: [Question regarding the specificity of the Cdc25 pT48 antibody]

The specific catalog numbers for the antibodies have now been provided, alleviating this concern.

>Reviewer#3.19: Also, Fig 9D-F do not have untreated controls (all samples have been treated with at least one drug). Do the cells have to grow in the presence of the drug to maintain the resistance phenotype?

>Authors' response R19: The resistant cells used in this experiment are constantly maintained in the presence of drugs. Therefore, a non-treated condition is not necessary. According to the legend, the data presented in Figure 9D and E were from "PC3 cells after transfection with siSCR or siCLU followed by treatment with cabazitaxel," not resistant cell lines.

13. The figure 9D legend was indeed referring to PC3 resistant cell lines and was corrected in the manuscript.

Other issues

The correlation coefficient value in the legend to Fig EV.1B: "In silico correlation analysis of Cdc25C and CLU mRNA levels in 460 prostate cancer patients using GeneSapiens data set. (Spearman correlation: rho -0.86; p-value = 0.0107)" do not match the values described in the text on page 4, "(p<0.001; r = -0.223)."

14. This mismatch of values was corrected. in Fig. EV.1B legend

> A description of the GeneSapiens resource has not been provided. A simple reference to the URL of the resource would be very useful.

15. A reference that describes the data set and the URL for GeneSapiens database has been added in the manuscript.

>Spelling error on page 13: Fiftheen days later

The spelling has been corrected

YOU MUST COMPLETE ALL CELLS WITH A PINK BACKGROUND ↓
 PLEASE NOTE THAT THIS CHECKLIST WILL BE PUBLISHED ALONGSIDE YOUR PAPER

Corresponding Author Name: Martin Gleave
Journal Submitted to: EMBO Molecular Medicine
Manuscript Number: : EMM-2015-06059